# HUMAN OR MACHINE? A PRELIMINARY TURING TEST FOR SPEECH-TO-SPEECH INTERACTION

**Xiang Li**[1,2,3,4*], **Jiabao Gao**[3*], **Sipei Lin**[3*], **Xuan Zhou**[3], **Chi Zhang**[3],
**Bo Cheng**[1], **Jiale Han**[5‡], **Benyou Wang**[2,3,4]

[1]State Key Laboratory of Networking and Switching Technology,
Beijing University of Posts and Telecommunications
[2]Shenzhen Research Institute of Big Data
[3]The Chinese University of Hong Kong, Shenzhen
[4]Shenzhen Loop Area Institute
[5]The Hong Kong University of Science and Technology
{lixiang2022,chengbo}@bupt.edu.cn,
{jiabaogao,sipeilin,xuanzhou,122090728}@link.cuhk.edu.cn,
jialehan@ust.hk, wangbenyou@cuhk.edu.cn

## ABSTRACT

The pursuit of human-like conversational agents has long been guided by the Turing test. For modern speech-to-speech (S2S) systems, a critical yet unanswered question is whether they can converse like humans. To tackle this, we conduct the first Turing test for S2S systems, collecting 2,968 human judgments on dialogues between 9 state-of-the-art S2S systems and 28 human participants. Our results deliver a clear finding: no existing evaluated S2S system passes the test, revealing a significant gap in human-likeness. To diagnose this failure, we develop a fine-grained taxonomy of 18 human-likeness dimensions and crowd-annotate our collected dialogues accordingly. Our analysis shows that the bottleneck is not semantic understanding but stems from paralinguistic features, emotional expressivity, and conversational persona. Furthermore, we find that off-the-shelf AI models perform unreliably as Turing test judges. In response, we propose an interpretable model that leverages the fine-grained human-likeness ratings and delivers accurate and transparent human-vs-machine discrimination, offering a powerful tool for automatic human-likeness evaluation. Our work[1] establishes the first human-likeness evaluation for S2S systems and moves beyond binary outcomes to enable detailed diagnostic insights, paving the way for human-like improvements in conversational AI systems.

## 1 INTRODUCTION

With the rapid advancement of generative artificial intelligence, large language models (OpenAI, 2023; Touvron et al., 2023; GLM et al., 2024) have become deeply integrated into people's daily lives, providing intelligent services through text-based human-machine interaction. As users seek more direct, hands-free, and immersive experiences, Speech-to-Speech (S2S) systems (ByteDance, 2025; Comanici et al., 2025) are gaining increasing attention by enabling interaction through the primary channel of human communication—*speech*. Such systems have broad applications, including empathetic social companions (Geng et al., 2025), personalized education (Galbraith & i Martínez, 2023), and interactive virtual assistants (TG et al., 2024). As the capabilities of S2S systems grow, a fundamental question emerges: do these systems converse like humans? Meeting this bar is strictly harder than text-based interaction, as it requires the models not only to achieve accurate semantic understanding and human-like persona alignment but also to ensure acoustic fidelity and emotional expression.

---

[*]Equal contribution. [‡]Corresponding author.
[1]We released code, data, and models at https://github.com/Carbohydrate1001/Turing-Test.

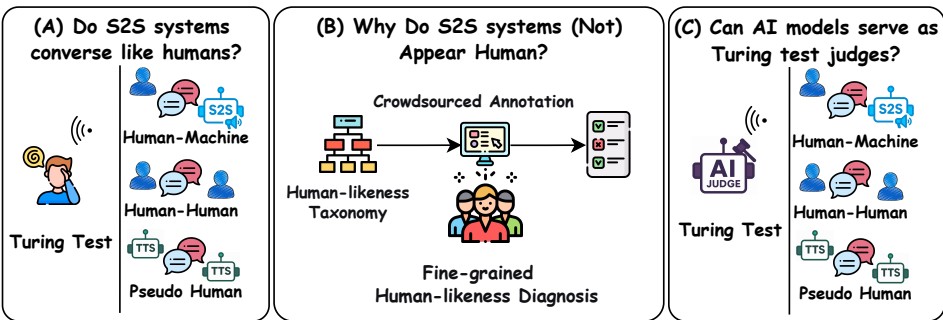

Figure 1: The design of our study.

In this work, we first investigate the human-likeness of current S2S systems by conducting Turing test. To facilitate this evaluation, we construct a high-quality dialogue dataset comprising human–human, human–machine, and pseudo-human (text-to-speech, TTS) dialogues. All human–machine dialogues are recorded in a professional studio with recruited volunteers. The dataset covers two languages, 10 topics, 9 state-of-the-art S2S systems, and 28 human speakers. We then deploy a gamified online platform to run the Turing test, collecting 2,968 judgments from 397 participants. Our results lead to a clear finding: no existing evaluated S2S models passes the Turing test, underscoring a substantial gap between current systems and truly human-like spoken interaction.

To move beyond a simple pass or fail outcome and understand the why behind this failure, we develop a fine-grained human-likeness taxonomy with 18 dimensions across five categories: semantic and pragmatic habits (Bottazzi Grifoni & Ferrario, 2025), non-physiological paralinguistic features (Warren et al., 2025), physiological paralinguistic features (Onda et al., 2025), mechanical persona (Fanous et al., 2025), and emotional expression (Wang et al., 2025a). By annotating our dialogue data accordingly, we diagnose the specific weaknesses of current S2S systems. Our analysis reveals that the artificial quality of current systems does not primarily stem from semantic deficiencies—in fact, contextual understanding is no longer the primary bottleneck, with models scoring near human levels on logical coherence and memory consistency. Instead, failures arise from deficiencies in paralinguistic features, emotional expression, and conversational persona. These findings collectively offer a concrete roadmap for developing more human-like S2S systems.

Finally, we explore the potential of automating the Turing test by asking: Can AI serve as the judge? We first demonstrate that 9 off-the-shelf AI models perform poorly at this task, failing to reliably distinguish human from machine-generated speech. In response, we develop a specialized and interpretable AI judge. Concretely, the model learns to score dialogues across the 18 human-likeness dimensions to capture fine-grained perceptual patterns. These interpretable scores are then fed into a regularized linear classifier to produce a final and explainable human–machine discrimination decision. This approach not only achieves strong performance but also provides transparent rationale for its judgments by linking them to specific human-likeness attributes. The resulting model offers a practical tool for diagnosing human-likeness of S2S systems with both headline scores and fine-grained attributions, thereby empowering rapid iteration toward more human-like systems.

An overview of our study design is shown in Figure 1. In summary, our work contributes (1) the first human- likeness evaluation on the current S2S systems via Turing test, (2) a comprehensive diagnostic framework and in-depth analysis explaining the gap in human-likeness, and (3) an effective and interpretable AI judge to automate human-likeness evaluation. Our code, dataset, and model are publicly available to foster progress in building truly human-like spoken dialogue agents.

## 2 BACKGROUND

**Turing Test** Since its introduction in 1950, the Turing Test (TURING, 1950) has served as a cornerstone for evaluating machine intelligence. Rathi et al. (2024) employ two variants of the Turing Test, the *Displaced Turing Test* and the *Inverted Turing Test*, to examine how well humans and large

Table 1: Existing Turing tests for AI.

| Turing Test | Modality |
|---|---|
| Jones & Bergen (2024a) | Text |
| Jones et al. (2025) | Text |
| Rathi et al. (2024) | Text |
| Chan (2003) | Text-Speech |
| Wang et al. (2025b) | Text-Speech |
| Ours | Speech-Speech |

language models can discriminate between online human–machine conversations, thereby reflecting the models' conversational perception abilities. Similarly, Jones & Bergen (2024a); Jones et al. (2025); Jones & Bergen (2025) design settings in which language models masquerade as humans in Turing Test scenarios to assess their linguistic expressiveness and emotional characteristics. In addition, Chan (2003); Wang et al. (2025b) extend the Turing Test paradigm to the domain of speech synthesis, evaluating the gap between synthetic speech and human dialogue to provide insights for model optimization. Inspired by these studies, we consider whether the Turing Test paradigm can be leveraged to evaluate speech-to-speech (S2S) systems, which constitute an indispensable component of contemporary human–machine interaction.

**Evaluation for S2S Systems**   Current evaluations of speech-to-speech (S2S) systems primarily focus on two dimensions: audio understanding and conversational intelligence. For example, Du et al. (2025) construct a multi-turn dialogue benchmark to assess pronunciation accuracy and the appropriateness of emotional expression in S2S systems. Jiang et al. (2025) propose an arena-style evaluation to measure instruction-following performance and paralinguistic expressiveness. Lin et al. (2025) assess dialogue fluency by analyzing response latency. In addition, Sakshi et al. (2024); Kumar et al. (2025b) design a suite of tasks such as speaker identification and emotion recognition to evaluate models' reasoning capabilities. More recent benchmarks such as VoiceBench (Chen et al., 2024) and MMAU-Pro (Kumar et al., 2025a) further expand the scope of evaluation: VoiceBench focuses on speech understanding in LLM-based voice assistants, while MMAU-Pro evaluates holistic audio understanding of multimodal AI models across speech, music, and general sound. Despite these advances, existing benchmarks differ fundamentally from our setting in both evaluation goal and evaluated modality. As summarized in Appendix A, prior work mainly measures whether models can correctly understand audio inputs or solve reasoning tasks, typically with text as the final output. In contrast, our work evaluates the *human-likeness* of S2S systems in multi-turn spoken interaction, where both input and output are speech. This distinction is important because success on conventional intelligence-oriented benchmarks does not necessarily imply that a system behaves in a more human-like manner.

## 3   DATASET CONSTRUCTION FOR THE S2S TURING TEST

We construct a dialogue dataset to support a rigorous and balanced evaluation of human-likeness in S2S systems. The dataset contains three categories of dialogues: human–machine (H-M), human–human (H-H), and pseudo human (PH). The following subsections detail the construction process.

### 3.1   HUMAN–MACHINE DIALOGUE

**Topic Design**   To ensure that the constructed human–machine dialogues are both authentic and diverse, we define 10 dialogue topics guided by DailyDialog (Li et al., 2017), which span a broad spectrum from daily life to financial activities. The detailed topics and their distribution in the final dialogues are illustrated in Figure 2.

**Model Selection**   In our experiments, we select 9 state-of-the-art S2S systems, spanning both open- and closed-source models, for human–machine dialogue generation. These include GPT-4o (Hurst et al., 2024), Gemini2.5-Pro (Comanici et al., 2025), Qwen3 (Yang et al., 2025), Kimi-K1.5 (Team et al., 2025b), ChatGLM-4.5 (Zeng et al., 2025), Hunyuan-TurboS (Team et al., 2025c), Doubao-Pro 1.5 (ByteDance, 2025), Claude-Sonnet 4 (Anthropic, 2024), and iFLYTEK-Spark (iFlytek, 2024). The detailed information about these models can be found in B.1.

**Dialogue Recording**   We invite 28 participants from 10 countries and regions to record human–machine dialogues in a professional recording studio, as detailed in Appendix B.2. Given a topic and a S2S system, the speaker is instructed to initiate and sustain a multi-turn conversation naturally around the given topic with the model, with the whole dialogue typically lasting between 20 to 60 seconds. Our goal is to elicit dialogues that are as human-like and realistic as possible. However, pilot runs revealed two key issues: (i) **identity disclosure**, S2S systems often proactively mention that they are intelligent assistants, which undermine the premise of Turing test, and (ii) **role passivity**, without contextual scaffolding, models fail to actively embody expected roles, instead

from a generic AI-assistant stance. To address these issues, we design three interaction strategies aimed at reducing identity leakage and encouraging immersive role-playing:

- **Human-Guided Initiation**. We let human speakers start the conversation by expressing opinions on an object or phenomenon, thereby preemptively suppressing the model's tendency to position itself as an assistant and setting a person-to-person tone. An example is `I always take a shower in the evening. I don't understand why there are people taking a shower in the morning.`

- **Role Playing**. In this setting, we assign the S2S system a concrete human role and background information via prompt, while explicitly instructing it not to disclose its identity. An example prompt is `You are now my mom and we are discussing my final exam grade. Please don't mention your identity in the subsequent conversation. Let's start chatting now.` The procedure is implemented as follows: we first have a test facilitator read the prompt to the S2S system to set the role and context, following which the recording start and the human speaker engage the model in conversation.

- **Human-Likeness Prompting**. To elicit more human-like conversational behavior from S2S systems, we augment the prompt with explicit instructions for human-like expression. This approach aligns with techniques used to enhance anthropomorphic behavior in large language models (Jones & Bergen, 2024b). As an illustration: `You are now my friend who came back from a vacation in Europe. Make your expression more humanlike. Don't mention your identity in the subsequent conversation. Let's start chatting now. How's your vacation to Europe?`

For a fair comparative evaluation of S2S systems, all participants are instructed to begin the dialogue with an identical initial opening utterance when engaging each S2S system. The specific utterances and prompts used are detailed in Appendix B.3. Finally, we perform manual filtering to remove dialogues in which the S2S system explicitly disclose its identity, respond in a non-target language, or exhibited overtly aggressive behavior during the interaction.

### 3.2 HUMAN–HUMAN DIALOGUE

To support comparative evaluation, we construct a human–human subset matched in scale and topic distribution to the human–machine subset, using a two-pronged approach: (i) Curated from existing datasets. We manually select dialogues from three open-source datasets DAILYTALK (Lee et al., 2023), IEMOCAP (Busso et al., 2008), and MagicData (Yang et al., 2022) that align with our predefined topics. During review, we observe frequent mutual interruptions that many S2S systems cannot yet emulate. To eliminate evaluation bias caused by this phenomenon, we filter out a considerable portion of dialogues with interruptions. In addition, to align with the alternating role patterns typical in human–machine multi-turn dialogues, we filter out dialogues with imbalanced participation from each speaker based on their engagement. Detailed settings can be found in the Appendix B.4. (ii) Recordings with volunteers. To ensure contextual consistency with the human–machine dialogues, we conduct an additional set of human–human recordings. In particular, we used the same opening utterances as those employed in the human–machine setup so as to maintain the same conversational topics and scenarios, thereby minimizing bias introduced by content differences.

### 3.3 PSEUDO HUMAN DIALOGUE SYNTHESIS

We notice that modern text-to-speech (TTS) models can synthesize dialogues with striking human-likeness. To raise the difficulty of Turing test, we introduce the dataset with pseudo-human dialogues synthesized by two state-of-the-art TTS models, Nari Dia-1.6B (nari-labs, 2025) and Spark-TTS (Wang et al., 2025c).We prepare scripts from two sources for TTS synthesis. First, we use a slightly modified version of the human-human dialogue script. Second, we prompt GPT-4o to generate two-speaker scripts conditioned on the predefined topics. Each utterance in the scripts is converted into speech using TTS models. Finally, we merge them into dialogues with a 180-230 ms inter-turn interval and add background ambience from reference recordings to enhance naturalness. The details on pseudo human dialogue synthesis are provided in Appendix B.5.

### 3.4 FINAL DATASET PROCESSING AND STATISTICS

For the collected dialogue data, we implement two bias-correction measures. First, we align the time intervals between both parties in the dialogues to avoid significant discrepancies in human subjective perception caused by overly long or short pauses, and to eliminate the impact of network latency or recording irregularities. Second, we bal-

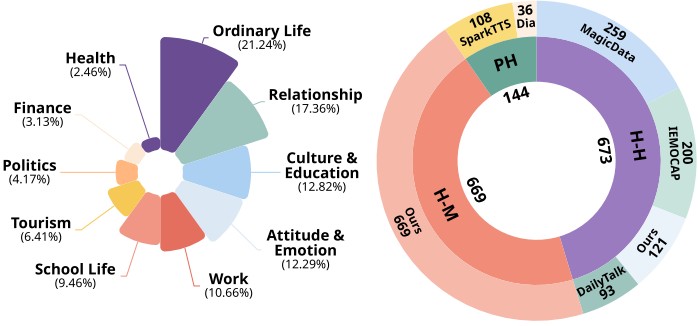

Figure 2: Data distribution.

ance the audio volume levels of both parties to ensure consistency, minimizing quality discrepancies introduced during the recording process.

The final dataset comprises a total of 1,486 dialogues, with a duration of 17.7 hours. This includes 669 human–machine dialogues (8.9 hours), 673 human–human dialogues (7.6 hours), and 144 pseudo-human dialogues (1.2 hours). The overall statistics are illustrated in Figure 2. We further divide the dataset into training and test sets, with the training set containing 525 human–machine and 531 human–human dialogues, totaling approximately 13.1 hours. The test set consists of 430 dialogues and 4.7 hours in total.

## 4 DO S2S SYSTEMS CONVERSE LIKE HUMANS?

**Game Platform Design for the Turing Test**   We deploy the Turing test as a lightweight and shareable game to encourage broad participation. Before playing, users complete a short questionnaire (age, gender, education, AI familiarity) and select their evaluation language (Chinese or English) to ensure judgments in their preferred language. In each round, users are required to evaluate a set of five dialogues. After listening to each dialogue, they determine whether Speaker B is human or machine. To boost engagement, participants receive points based on the accuracy of their judgments, and a public leaderboard ranks all players based on their performance. A built-in sharing feature helps disseminate the game to a wider audience, facilitating larger-scale data collection. The main interface appears in Figure 3, with details in Appendix C.1. By September 15, 2025, the platform has collected results from 397 participants, totaling 2,968 dialogue evaluations. Our game platform supports long-term and scalable Turing test.

**Turing Test Results and Analysis**   Our evaluation employs the *Success Rate* as the primary metric for assessing human-likeness, which reflects the proportion of trials in which a system is judged to be human by evaluators. A value greater than 0.5 would suggest that human evaluators are incapable of distinguishing the model from a human (Jones & Bergen, 2024b). We also examine participant *Accuracy* across different demographic groups, defined as the proportion of correct human-versus-machine identifications. This allows us to investigate how factors such as age, gender, education, and AI familiarity influence human perceptual bias in the Turing test.

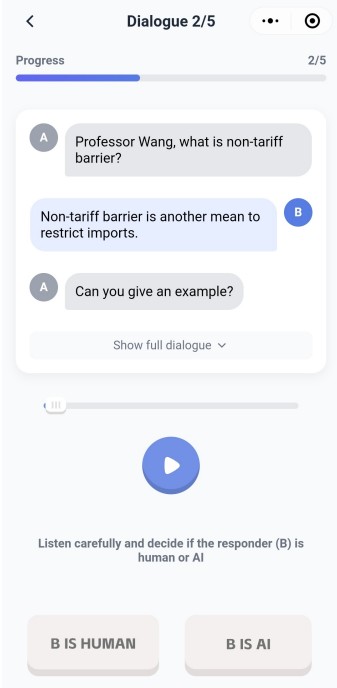

Figure 3: The main game interface of the Turing test.

> **Observation 1:** *No existing evaluated S2S system passes the Turing test.*

As shown in Figure 4a, human-to-human dialogues achieve success rates as high as 0.87 for English and 0.70 for Chinese, confirming the robustness of our evaluation design. In contrast, all S2S systems perform significantly below the 0.5 chance threshold, with success rates ranging from 0.07

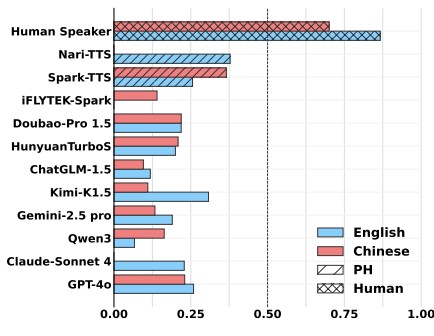

(a) Success rate across S2S systems.

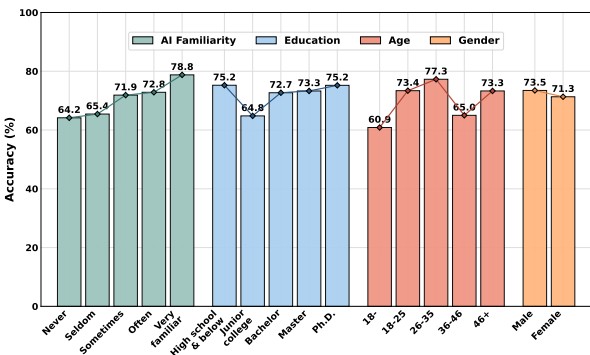

(b) Accuracy across different groups.

Figure 4: (a) Turing test success rates of S2S systems, measured as the proportion of responses judged as human. Higher values indicate greater human-likeness. (b) Participant accuracy in identifying human vs. machine. Detailed scores and results categorized by interaction strategies are provided in Appendix C.2.

to 0.31. This significant performance gap highlights the fundamental limitations of current speech models in their ability to simulate human-like behavior. Moreover, the success rates for pseudo human dialogues fall short of human-to-human performance, suggesting that even when scripts are highly similar to real conversations, synthesized speech still lacks sufficient acoustic naturalness to pass as humans. However, despite sharing similar limitations in vocal quality, pseudo human dialogues still surpass most S2S systems, suggesting that current S2S systems are bottlenecked not only by speech synthesis quality, but also by higher-level vocal interaction capabilities such as speech understanding, role-based acoustic adherence, and conversational reasoning.

> **Observation 2:** *An individual's ability to distinguish humans from machines depends more on experience than on demographics.*

As shown in Figure 4b, participants with greater AI familiarity achieve clearly higher detection accuracy, reaching 78.8% for the most experienced group versus 64.2% for the least familiar group. Younger cohorts also outperform older groups, likely due to more frequent exposure to AI interactions and heightened sensitivity to non-human cues. In contrast, accuracy shows minimal variation by gender or education level. These results suggest that detection ability is shaped more by experiential factors than demographic traits. As public familiarity with AI grows, passing Turing tests may become progressively harder over time. *Our game-based evaluation platform supports longitudinal Turing testing and periodic recalibration*, enabling continued assessment of human-likeness against evolving human judgment standards.

## 5 WHY DO S2S SYSTEMS (NOT) APPEAR HUMAN?

To systematically investigate *why* current S2S systems fail to pass as human, we develop a comprehensive taxonomy for human-likeness diagnosis comprising five major categories and 18 fine-grained dimensions (see Appendix D.1). Using this taxonomy, all dialogue samples are crowd-sourced and rated on a 5-point rating scale (Appendix D.2), after which human experts reviewed and refined the labels to ensure quality (Appendix D.3). The resulting labels enable a granular diagnosis of failure modes that limit the human-likeness of current speech models. As illustrated in Figure 5, we summarize four key observations that explain the pros and cons of current S2S systems in achieving human-like naturalness, therefore providing guidance for developing advanced and human-like S2S systems.

> **Observation 3:** *Semantic and contextual understanding in dialogues are not the primary bottlenecks for S2S systems.*

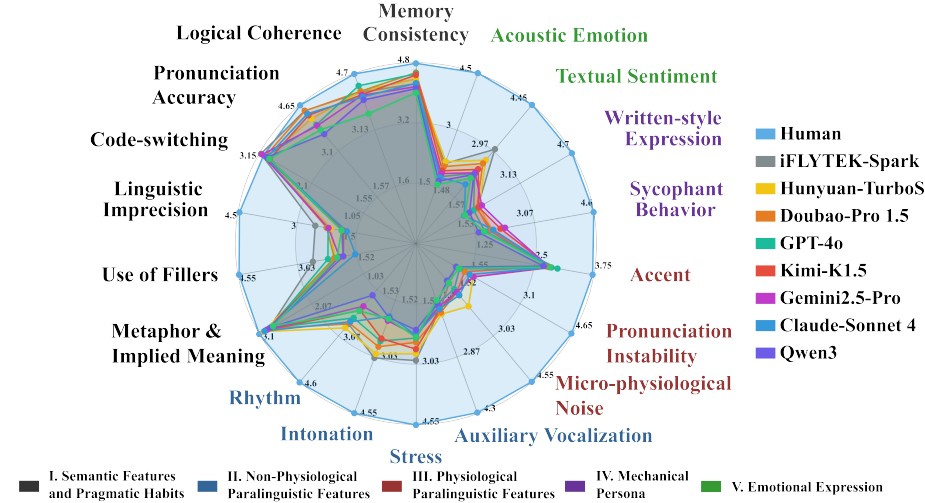

Figure 5: Crowd-annotated scores (1–5) across the 18 human-likeness dimensions.

Current models demonstrate remarkable proficiency in core semantic tasks, closely approaching human-level performance. Specifically, models excel in Memory Consistency, capably retaining and referencing information within a short dialogue context, and in Logical Coherence, ensuring smooth transitions between turns without abrupt contradictions. Furthermore, Pronunciation Accuracy is generally high, with modern systems correctly articulating words, including challenging heteronyms. These strengths indicate that S2S systems have largely solved the foundational challenges of textual understanding and generating clear and coherent dialogue scripts.

**Observation 4:** *The speech generated by S2S systems often lacks human-like paralinguistic features, exhibiting rigid prosody and absence of disfluency cues.*

Across non-physiological paralinguistic features, S2S outputs show pronounced deficits in vocal dynamics. Rhythm and intonation changes are mechanically regular, with few context-appropriate pauses or pitch movements. Stress on salient words is weak or misplaced, which is a crucial element of human communication. Furthermore, models avoid human disfluency cues, such as linguistic imprecision (e.g., hedges like "probably"), use of fillers ("um"), and micro-physiological noises (e.g., breath sounds). These paralinguistic shortcomings, even when the content is fluent, make the speaker perceptibly machine-like.

**Observation 5:** *Emotional expressivity remains largely limited in current S2S systems.*

The textual sentiment scores of S2S systems are significantly lower than human performance, reflecting the lack of nuanced emotions due to the writing-style expressions. More critically, the acoustic emotion scores are even lower than those of textual sentiment, due to rigid prosody and weak or misaligned stress patterns. This indicates that S2S systems tend to generate dialogues with neutral and unconvincing emotional tones, making them readily perceived as non-human by listeners.

**Observation 6:** *The persona of S2S systems is often perceived as mechanical, characterized by excessively sycophantic and formal expression.*

S2S systems reveal a mechanical persona through their social interaction. Unlike humans who judiciously agree or disagree based on context, current models exhibit a strong default tendency to excessively affirm, apologize, and express gratitude. For instance, to a user's statement like, "*I'm planning to go around in Korea for 5 days*", a model might respond with disproportionate enthusiasm such as, "*That's absolutely amazing—fantastic choice!*". Moreover, their written-style expression skews formal, lacking the conversational looseness typical of spontaneous speech.

# 6 CAN AI MODELS SERVE AS TURING TEST JUDGES?

## 6.1 TURING TEST WITH AI JUDGES

To explore whether AI models can reliably assess human-likeness in dialogues, we employ 9 state-of-the-art models as automated judges, and each model is tasked with classifying whether a given dialogue response is human- or machine-generated. Detailed prompts are provided in Appendix E.1. Table 2 reports their classification accuracy across the three dialogue types (human–human, human–machine, and pseudo human).

Table 2: AI judge accuracy of different models on the Turing test data.

| Model | ACC(H-H)↑ | ACC(H-M)↑ | ACC(PH)↑ | Overall↑ |
|---|---|---|---|---|
| Human Judgement | 0.7028 | 0.8357 | 0.6384 | **0.7284** |
| Baichuan-Audio(Li et al., 2025) | 0.8169 | 0.1528 | 0.1250 | 0.3628 |
| Gemini 2.5 pro(Comanici et al., 2025) | 0.5775 | 0.7292 | 0.5764 | 0.6279 |
| Gemma 3n(Team et al., 2025a) | 0.4648 | 0.4444 | 0.4028 | 0.4372 |
| GPT-4o-Audio-Preview(Hurst et al., 2024) | **0.9648** | 0.2708 | 0.0069 | 0.4116 |
| MiniCPM-o 2.6(Yao et al., 2024) | 0.6761 | 0.4306 | 0.2986 | 0.4674 |
| Phi-4-Multimodal(Abouelenin et al., 2025) | 0.7746 | 0.1458 | 0.2222 | 0.3791 |
| Seallms-Audio(Nguyen et al., 2023) | 0.1127 | **0.8472** | **0.7292** | 0.5651 |
| Voxtral Mini(Li et al., 2025) | 0.5141 | 0.5069 | 0.3889 | 0.4698 |
| Qwen2.5-Omni(Xu et al., 2025) | 0.7817 | 0.2361 | 0.2361 | 0.4163 |
| Average of Model Judgement | 0.6238 | 0.4011 | 0.3130 | 0.4527 |

**Observation 7:** *Existing AI judges significantly underperform humans in the Turing test and exhibit systematic bias.*

The overall performance of the AI judges (average accuracy: 0.4527) remains substantially lower than that of human evaluators (accuracy: 0.7284), with even the best-performing model Gemini 2.5 Pro achieving only 0.6279 accuracy. Analysis of model behavior reveals three distinct bias patterns: several models (e.g., GPT-4o-Audio-Preview, Baichuan-Audio, Phi-4-Multimodal, Qwen2.5-Omni) exhibit a strong tendency to classify most dialogues as human–human, models such as SeaLLMs-Audio display the opposite bias toward human–machine judgments, while Voxtral Mini behaves close to random guessing. These results highlight the current limitations of multimodal models in replicating human-like perceptual judgment in Turing test scenarios.

## 6.2 INTERPRETABLE AI JUDGE FOR HUMAN-LIKENESS EVALUATION

Given that general-purpose large models perform unreliably as human-likeness judges, we develop an interpretable multimodal evaluator designed to deliver transparent and trustworthy decisions. Detailed experimental setup is provided in Appendix E.2.

### 6.2.1 TRAINING FRAMEWORK

We adopt a two-stage fine-tuning framework on Qwen2.5-Omni, which trains the model to first capture fine-grained human-likeness patterns and then produce a final and explainable human–machine discrimination decision.

**Fine-grained Scoring Projection.** Given an audio dialogue $x \in \mathcal{D}$, we first encode it with a pre-trained audio–language model (ALM) to obtain a fixed-dimensional representation $h = f_{\text{ALM}}(x) \in \mathbb{R}^d$ (a two-source fused pooling, see Appendix E.3 for representation design). We then map $h$ to *interpretable* dimension scores with an Ordinal Discretization Layer (ODL) (Tutz, 2022):

$$z = f_{\text{ODL}}(h;\theta) \in \mathbb{R}^K, \qquad z_k = [f_{\text{ODL}}(h;\theta)]_k$$

where $K$ is the number of fine-grained human-likeness dimensions and $z_k$ is the latent score for dimension $k$. To respect the ordinal nature of human ratings (e.g., $r$ ordered levels, 1–$r$), we convert

each $z_k$ into an ordinal distribution via *ordered cut-points*. For each dimension $k \in \{1, \ldots, K\}$, we define $r - 1$ strictly ordered cut-points

$$C_{ik} = \frac{i - r + 2}{2(r - 2)} s_k, \qquad i \in \{1, \ldots, r - 1\}$$

where $s_k$ is a learnable scale that controls bin spacing. Using a cumulative-link formulation, cumulative probabilities are

$$P(Y_k \leq i \mid x) = \sigma(C_{ik} - z_k),$$

where $\sigma(\cdot)$ denotes the sigmoid function. Per-category probabilities follow by differencing: $P(Y_k = 1) = P(Y_k \leq 1)$, $P(Y_k = i) = P(Y_k \leq i) - P(Y_k \leq i - 1)$ for $2 \leq i \leq r - 1$, and $P(Y_k = r) = 1 - P(Y_k \leq r - 1)$. Let $S_H(x) \in \{1, \ldots, r\}^K$ denote human-likeness ratings for $x$, we fit the ODL by minimizing the ordinal negative log-likelihood over all samples and dimensions:

$$\min_{\mathbf{s}, \theta} \frac{1}{|\mathcal{D}|} \sum_{x \in \mathcal{D}} \sum_{k=1}^{K} \left[ -\log P(Y_k = S_H^{(k)}(x) \mid x) \right]$$

This procedure yields $K$ order-preserving, human-aligned scores per dialogue that serve as interpretable inputs for the final human–vs.–machine classifier.

**Explainable Binary Classification.** After training the ODL, each of the $k$ neurons acquires an ordinally constrained scoring pattern induced by the cut-point scheme. Consequently, the ODL outputs are no longer arbitrary latent features; they instantiate interpretable scoring dimensions aligned with human ratings and preserve their ordinal structure for human–machine discrimination. Leveraging this property, we feed the logits $z$ into a linear classifier with regularization constraint to ensure that the final classification remains interpretable:

$$\min_{W_F} \frac{1}{|\mathcal{D}|} \sum_{(x,y) \in \mathcal{D}} \mathcal{L}_{\mathrm{CE}}(W_F z, y) + \lambda R(W_F)$$

where $\mathcal{L}_{\mathrm{CE}}$ is the Cross-Entropy Loss, $W_F \in \mathbb{R}^{n \times K}$ is the weight matrix of the final linear layer with $n$ categories, $y$ is the label of $x$, $R(W) = ||W_1 + W_2||_2$ is the symmetry regularization, and $\lambda$ is set to 0.1. Model ablations and hyperparameter tuning details are provided in Appendix E.4 and E.5.

### 6.2.2 RESULTS AND DISCUSSION

Table 3: Binary classification accuracy of different models across three evaluation data types.

| Data Type | Qwen2.5-Omni | Qwen2.5-Omni(LoRA) | Human Judge | Ours |
|---|---|---|---|---|
| Human-Human↑ | 0.7817 | 0.9230 | 0.7028 | **0.9507** |
| Human-Machine↑ | 0.2361 | 0.6319 | 0.8357 | **0.9722** |
| Pseudo Human ↑ | 0.2361 | 0.0972 | 0.6384 | **0.9306** |
| Overall ↑ | 0.4163 | 0.5744 | 0.7284 | **0.9605** |

We evaluate the interpretable AI judge on the Turing test using binary classification accuracy (human vs. machine). As presented in Table 3, Qwen2.5-Omni (LoRA) represents Qwen2.5-Omni fine-tuned using LoRA technology (Hu et al., 2022). It can be observed that our approach outperforms all variants and human evaluators. The overall accuracy is 23.21% higher than the human evaluation, 38.61% higher than the LoRA-based approach, and more than doubles the performance of the original model. Notably, the model achieves 93.06% accuracy on pseudo-human dialogues unseen during training, demonstrating strong generalization. In addition, the model shows strong consistency with fine-grained human ratings, a capability facilitated by its interpretable design (see Appendix E.6).

**Out-of-Domain Generalization Evaluation** We further evaluated our model on three out-of-domain (OOD) datasets that span diverse acoustic, demographic, and interaction conditions: 1) CosyVoice2 Synthesis (Pseudo Human) (Du et al., 2024), synthesized dialogues across different age groups (older adults and children); 2) Fisher (Human-Human) (Cieri et al., 2004), telephone speech with significant background noise; 3) MultiDialog (Human-Human) (Park et al., 2024): clean background native-speaker dialogue recordings. We sample 64 dialogues from each dataset for evaluation. In addition to accuracy, we introduced the ROC-AUC score to provide a robust and threshold-independent evaluation of classification performance. The results of human–machine classification are presented in Table 4. These results indicate that the model generalizes well and maintains stable performance under distribution shift.

Table 4: Binary classification accuracy and ROC-AUC on OOD test set.

| Metric | Overall (Inner) | CosyVoice2 | Fisher | MultiDialog | Overall (OOD) |
|---|---|---|---|---|---|
| Accuracy | 0.9605 | 0.9844 | 0.9844 | 0.9531 | 0.9740 |
| ROC-AUC | 0.9791 | – | – | – | 0.9881 |

**Observation 8:** *Our interpretable AI judge delivers superior performance in distinguishing human from machine-generated speech. By providing both an overall human-likeness score and fine-grained diagnostics, it serves as a practical tool for S2S assessment.*

## 7 CONCLUSION

This work presents the first Turing test for modern S2S systems, delivered via a game-based online platform that enables large-scale and longitudinal testing. Our findings reveal a clear gap: no current system passes, demonstrating that human-like conversational ability remains an unsolved challenge. Through an 18-dimension taxonomy, we show the bottleneck has shifted from semantic understanding to shortcomings in paralinguistic features, emotional expressivity, and conversational persona, explaining why even fluent S2S output sounds distinctly artificial. To support automatic evaluation, we develop an interpretable AI judge that significantly outperforms off-the-shelf models and provides diagnostic insights.

*Impact.* We provide the community with a new human-likeness evaluation framework for S2S systems and move beyond binary pass/fail to automatic, diagnostic, and scalable evaluation. Our results offer practical guidance toward more genuinely human-like S2S systems by identifying the core challenges in acoustic naturalness, emotional expressivity, and social behavior.

## ETHICS STATEMENT

Our study involves the collection of audio recordings from human participants. In conducting this research, we have adhered to strict ethical principles to safeguard participants' privacy, autonomy, and well-being. The main ethical considerations are outlined below:

- **Informed Consent:** All participants were clearly informed that their speech would be recorded and potentially used in academic publications. Participation was voluntary, and individuals had the right to withdraw at any stage without penalty.
- **Data Anonymization:** To ensure participant confidentiality, all audio recordings were anonymized by removing any personally identifiable information, making it impossible to trace the data back to individuals.
- **Data Security:** Collected data are stored under strict security protocols, with access limited to authorized research personnel. Comprehensive measures are in place to prevent unauthorized access, disclosure, or misuse.
- **Scientific Integrity:** We maintain high standards of transparency and accuracy in reporting methods and results. The research is presented in a manner that supports reproducibility, and all contributions are properly acknowledged.

- **Avoiding Harm and Promoting Fairness:** We have taken measures to minimize potential harm and avoid reinforcing social biases. Our work is committed to fairness, inclusivity, and respect for participants, with the goal that research outcomes be applied in a socially responsible manner.

We reiterate that all data and models are intended solely for scientific research purposes, and must not be used for commercial activities or any unlawful or fraudulent actions.

## REPRODUCIBILITY STATEMENT

We have made every effort to ensure that the results presented in this paper are reproducible. All code and datasets have been made publicly available in an anonymous repository to facilitate replication and verification.

Our experiments comprise three main components. First, we collected dialogue data for the Turing test, which includes human–machine dialogues (see Section 3.1 for the detailed procedure), human–human dialogues (Section 3.2), and pseudo human–human dialogues generated via TTS models (Text-to-Speech, also described in Section 3.3). Based on the collected data, we designed a game-based human evaluation platform supporting fine-grained annotation, with the detailed design and implementation process outlined in Section 4. Furthermore, we developed a fine-grained annotation protocol incorporating expert validation, as described in Appendix D.1. Using this protocol, we conducted crowd-sourced annotation; the design of the annotation platform is provided in Appendix D.2. Finally, we trained a human-like judge model using the annotated data, with the model training procedure and hyperparameter settings detailed in Appendix E. We believe that these comprehensive descriptions significantly enhance the reproducibility of our work.

## ACKNOWLEDGMENTS

This work is supported in part by Longgang District Special Funds for Science and Technology Innovation under Grant LGKCSDPT2023002; the National Natural Science Foundation of China under Grants 62372058, U22A2026.

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

## THE USE OF LARGE LANGUAGE MODELS

**Using an LLM to help with paper writing**  During the preparation of this work, the authors utilized Large Language Models for language polishing, improving the structural clarity of the manuscript, and refining the formal expression of individual sentences. The use of Large Language Models did not influence the substantive content of the study and served solely as a writing aid.

### OVERALL OF THE APPENDIX

The appendix provides supplementary material to support the methodology outlined in the main text. It is organized into four sections for clarity:

- Appendix A: **Comparison with Existing Speech Benchmarks** details the distinctions between our benchmark and existing speech benchmarks.
- Appendix B: **Data Collection** details the procedures, sources, and criteria used for gathering the raw data utilized in this study.
- Appendix C: **Turing-Test** describes the design of the human evaluation (Turing test).
- Appendix D: **Fine-Grained Human-Likeness Dimension Annotation** details the comprehensive guidelines followed for data annotation.
- Appendix E: **Training Details** specifies the key hyperparameters, computational environment, and training configurations of the models.

## A  COMPARISON WITH EXISTING SPEECH BENCHMARKS

We conducted a detailed comparison between our work and two representative speech benchmarks, VoiceBench (Chen et al., 2024) and MMAU-Pro (Kumar et al., 2025a). As summarized in Table 5, our work differs fundamentally in evaluation goal and evaluated modality.

Table 5: Comparison with Existing Speech Benchmarks.

| Aspect | VoiceBench | MMAU-Pro | Turing Test (Ours) |
|---|---|---|---|
| Goal | Evaluating speech understanding in LLM-based voice assistants | Evaluating holistic audio understanding of multimodal AI models across speech, music, and sound | Evaluating human-likeness of Speech-to-Speech systems |
| Input Modality | Speech or Text | Speech and Text | Speech |
| Output Modality | Text | Text | Speech |
| Dialogue Turns | Single-turn | Multi-turn | Multi-turn |
| The Smarter the Better? | Yes—higher intelligence implies better performance | Yes—higher intelligence implies better performance | No—being "too smart" does not necessarily make a model more likely to pass the Turing Test |

To further examine whether "being smarter" makes a model more human-like, we selected S2S systems that appear in both MMAU-Pro and our study, and compared their reasoning accuracy on MMAU-Pro with their Turing Test pass rates. The results are summarized in Table 6.

The Pearson correlation between reasoning accuracy and Turing test pass rate is 0.0456. This indicates that reasoning ability is nearly uncorrelated with human-likeness in current S2S systems,

Table 6: Reasoning Ability vs. Human-likeness in Speech-to-Speech Models.

| Model | Reasoning Accuracy (MMAU-Pro) | Turing Test Pass Rate (%) |
|---|---|---|
| Kimi-K1.5 | 46.6 | 12.7 |
| Qwen3 | 52.2 | 15.1 |
| GPT-4o | 52.5 | 23.0 |
| Gemini-2.5-Pro | 59.2 | 13.7 |

revealing a disconnect between traditional intelligence benchmarks and the human-likeness required for speech interaction.

# B  DATA COLLECTION

The section is organized into the following sections:

- Section B.1: Model Selection for the Turing Test.

- Section B.2: Participant Profiles.

- Section B.3: Human-Machine Dialogue Initialization Design Details.

- Section B.4: Human-Human Dialogue Filtering.

- Section B.5: Pseudo Human Dialogue Synthesis.

## B.1  MODEL SELECTION FOR THE TURING TEST

All S2S Systems we selected for evaluation are shown in Table 7. During pilot recordings and testing, we observe that Claude-Sonnet 4 supports only English conversations, while iFLYTEK-Spark exhibits suboptimal performance on long English prompts due to its underlying training constraints. To ensure dialogue quality, we generate dialogues in English for Claude-Sonnet 4 and in Chinese for iFLYTEK-Spark.

Table 7: Models used for the Turing test.

| Model | Release Year | Open-Source | # Dialogues | Share (%) | Language |
|---|---|---|---|---|---|
| GPT-4o (Hurst et al., 2024) | 2024 | × | 89 | 13.30% | CN & EN |
| Gemini2.5-Pro (Comanici et al., 2025) | 2025 | × | 82 | 12.26% | CN & EN |
| Qwen3 (Yang et al., 2025) | 2025 | ✓ | 83 | 12.41% | CN & EN |
| Kimi-K1.5 (Team et al., 2025b) | 2025 | × | 83 | 12.41% | CN & EN |
| ChatGLM-4.5 (Zeng et al., 2025) | 2025 | ✓ | 77 | 11.51% | CN & EN |
| Hunyuan-TurboS (Team et al., 2025c) | 2025 | × | 86 | 12.86% | CN & EN |
| Doubao-Pro 1.5 (ByteDance, 2025) | 2025 | × | 85 | 12.71% | CN & EN |
| Claude-Sonnet 4 (Anthropic, 2024) | 2025 | × | 41 | 6.13% | EN |
| iFLYTEK-Spark (iFlytek, 2024) | 2025 | × | 43 | 6.43% | CN |

## B.2  PARTICIPANT PROFILES

We provide the detailed profiles of all 28 participants in Table 8.

Table 8: Participant Profiles.

| Speaker ID | Chinese | English | Country / Region |
|---|---|---|---|
| speaker01 | ✓ | × | China |
| speaker02 | ✓ | ✓ | China |
| speaker03 | ✓ | × | China |
| speaker04 | ✓ | × | China |
| speaker05 | ✓ | ✓ | China |
| speaker06 | ✓ | ✓ | China |
| speaker07 | ✓ | × | China |
| speaker08 | × | ✓ | China |
| speaker09 | ✓ | × | China |
| speaker10 | ✓ | × | China |
| speaker11 | ✓ | × | China |
| speaker12 | ✓ | × | China |
| speaker13 | ✓ | ✓ | Hong Kong, China |
| speaker14 | × | ✓ | Pakistan |
| speaker15 | × | ✓ | Tajikistan |
| speaker16 | × | ✓ | Malaysia |
| speaker17 | × | ✓ | Indonesia |
| speaker18 | × | ✓ | Russia |
| speaker19 | × | ✓ | Indonesia |
| speaker20 | × | ✓ | Greece |
| speaker21 | × | ✓ | Indonesia |
| speaker22 | × | ✓ | Indonesia |
| speaker23 | × | ✓ | UK |
| speaker24 | × | ✓ | US |
| speaker25 | × | ✓ | Indonesia |
| speaker26 | ✓ | × | China |
| speaker27 | ✓ | × | China |
| speaker28 | × | ✓ | Indonesia |

### B.3 Human-Machine Dialogue Initialization Design Details

For the Turing evaluation, we collect 2 *Human-Guided Initiation* (Figure 6), 3 *Role Playing* (Figure 7), and 4 *Human-Likeness Prompting* (Figure 8) initialization evaluation dialogues for both English and Chinese (if applicable) from each S2S system. For any of the specific initialization, we fixed the starting sentences that interact with these 9 models. Eventually, we obtained 144 human-machine data for evaluation in total. The reason for including more dialogues for *Role Playing* than *Human-Guided Initiation* is that, the former one tend to leads the conversation to discussion on viewpoints. This phenomenon limits the dialogue coverage to only a narrow range of everyday scenarios. Thus, we limit the amount of *Human-Guided Initiation*. By contrast, we include more dialogues for *Human-Likeness Prompting* than *Role Playing* because *Human-Likeness Prompting* explicitly attempts to elicit stronger humanlike qualities from S2S systems. This design allows our dataset to capture a richer and more human-like spectrum of conversational behavior. The following figures show what the 18 dialogue initializations are.

### B.4 Human-human Dialogue Filtering

For the human-human dialogues, we extracted or recorded conversation segments of around 20–60 seconds to align with the human-machine dialogues. On one hand, too short dialogues may present little context. On the other hand, excessively long recordings are not available for some S2S system. To ensure balanced interactions, we retained only segments in which each of the two speakers contributed roughly equally, defined as having approximately 50% of the total utterances.

"到冬天就是真的不想去澡堂，太冷了。"

"诶，我上周末考的托福出分了，考的挺不好，感觉又浪费了两千块钱。"

*"I always take a shower in the evening. I don't understand why there are people taking a shower in the morning."*

*"I still get nervous before every test, no matter how prepared I am."*

Figure 6: Human-guided initiation (2 ZH 2 EN).

"你现在扮演一个一个超市售货员，我是一个来超市买物品的顾客，请你不要在后续聊天提及你的身份，我们现在开始聊天。帮我拿一斤苹果和桃子。"

"现在你扮演一个外卖员，我点了你要送的餐，你在跟我打电话，请你不要在后续聊天提及你的身份，我们现在开始聊天。你好，请问我的外卖什么时候到啊？"

"你现在扮演我的朋友，我邀请你来我家吃饭，请你不要在后续聊天中提及你的身份，我们现在开始聊天。今晚有啥想吃的？"

*"You are now a university student, and we are discussing about university ranking. Please don't mention your identity in the subsequent conversation. Let's start chatting now. Do you know that recently the student of our university has been in some conflict with the student of our brother university?"*

*"You are now my friend, and I invite you to my home for dinner tonight. Please don't mention your identity in the subsequent conversation. Let's start chatting now. What do you want to have for dinner tonight?"*

*"You are now my mom and we are discussing my final exam grade. Please don't mention your identity in the subsequent conversation. Let's start chatting now. Mom, I only scored 60 in my math exam."*

Figure 7: Role playing (3 ZH, 3 EN).

## B.5 PSEUDO HUMAN DIALOGUE SYNTHESIS

**Dialogue Scripts for TTS** The dialogue scripts cover 10 topics as our dataset. Each script presents a conversation between two speakers. We obtain scripts in two ways:

1. We use ChatGPT to adjust our existing dialogue scripts, ensuring that the original meaning remains intact while maintaining a natural, conversational tone. This part of the scripts contains all of the HH data in the additional set that ensures contextual consistency. This allows us to generate data that closely resembles our previous human-to-human dialogues. On the one hand, the scripts are grounded in authentic everyday conversations. On the other hand, the similarity in content helps reduce the chance that audiences distinguish between human and machine solely based on biases introduced by dialogue content . The prompt used in this way is shown as follow:

"请你扮演我的同事，我们在闲聊，请让你的表达尽可能地像人，尽量让我相信你是人类，请你不要在后续聊天提及你的身份，我们现在开始聊天。我们这个月的 *kpi* 快完不成了，老板给的压力太大了。"

"请你扮演我的女朋友，我们在一起散步，请让你的表达尽可能地像人，尽量让我相信你是人类，请你不要在后续聊天提及你的身份，我们现在开始聊天。我在工作上的时间太多了，最近没有能好好陪你。"

"请你扮演我的朋友，上周末你刚去过三亚旅游，请让你的表达尽可能地像人，尽量让我相信你是人类，请你不要在后续聊天提及你的身份，我们现在开始聊天。三亚好玩不？"

"现在你扮演呆在教室的同学，现在突然下雨了，我来找你借伞，请让你的表达尽可能地像人，尽量让我相信你是人类，请你不要在后续聊天提及你的身份，我们现在开始聊天。诶同学，请问一下你有多余的伞吗？突然下雨了，想借一下。"

*"You are now my friend who came back from a vacation in Europe. Make your expression more humanlike. Don't mention your identity in the subsequent conversation. Let's start chatting now. How's your vacation to Europe?"*

*"You are now my classmate who still stays in the classroom. Rain suddenly starts pouring outside. Make your expression more humanlike. Don't mention your identity in the subsequent conversation. Let's start chatting now. Excuse me⋯ Hey, do you happen to have an umbrella I could borrow?"*

*"You're now a taxi driver, and I'm a passenger in your cab. Make your expression more humanlike. Don't mention your identity in the subsequent conversation. Let's start chatting now. I am new here, can you take me to the best restaurant in town?"*

*"You are now my colleague who stayed late at the office with me to finish a deadline. Make your expression more humanlike. Don't mention your identity in the subsequent conversation. Let's start chatting now. I don't think we could get this done tonight."*

Figure 8: Human-likeness prompting (4 ZH, 4 EN).

You are a language refinement expert.
Without changing the original meaning or the overall flow of the dialogue, your task is to slightly adjust the conversation between two speakers. The goal is to preserve a natural, everyday tone. You may apply techniques such as: adding light interjections or filler words to make the speech sound more authentic, or rephrasing sentences into alternative but commonly used everyday expressions.
Please return only the refined dialogue in JSON format, keeping the same structure as the original.
Original dialogue:
{Utterances_in_json_format}
Adjusted dialogue:

2. We generate new 40–50 second everyday dialogue scripts with GPT-4o, based on given themes (topic 1–10). The prompt used in this way is shown as follow:

You are a writing expert.
Please generate a spoken-style dialogue script between two people on the topic of "{topic}."
Please follow these requirements: 1. The dialogue should sound natural, conversational, and realistic. 2. Add a small number of interjections (e.g., "ah," "oh," "hey") and filler words (e.g., "um," "you know," "like") to enhance authenticity. 3. The dialogue content should be logically coherent and reflect everyday life experiences. 4. The total length should correspond to 40–50 seconds of speaking time. 5. Use "A" and "B" as speaker labels. 6. The output format must be a JSON array, with the following structure only:

```
[
  {
    "speaker": "A",
    "text": "First utterance"
  },
  {
    "speaker": "B",
    "text": "Second utterance"
  }...
]
```
Please return only the JSON array.

**Audio Synthesis and Dialogues Merging**  For each dialogue script involving two speakers we attained, we selected the voices of two participants in human–machine dialogue recordings and performed voice cloning for each speaker's individual utterances. This approach helps mitigate bias caused by speaker voice characteristics, preventing users from inferring the human or machine identity of the responder based solely on speaker A's voice.

After generating the speech for each utterance from both sides, we concatenated them in dialogue order to form a complete conversation. Between each utterance, we inserted a random pause of 180–230 milliseconds to ensure natural timing between sentences. Finally, we added a short background noise sample, taken from the reference voice, over the entire concatenated dialogue to further enhance the naturalness of the conversation.

Through the above process, we obtained a complete pseudo-human dialogue. In total, 36 dialogues were generated with Nari-TTS (all English), and another 108 with Spark-TTS(36 English, 72 Chinese). Since Nari Dia-1.6B only supports English, it is used exclusively for English dialogues.

**Use of the Pseudo Human Dialogues**  All Pseudo human dialogues synthesized by TTS were included only in the Turing evaluation set and were not used for training our evaluator. These dialogues were incorporated into the gamified Turing Test released on social media, making the task more challenging and engaging. As shown in Figure 4a, TTS models achieved the highest success rate in the machine side.

## C   TURING-TEST

The section is organized into the following sections:

- Section C.1: Turing Test Platform.
- Section C.2: Supplementary Turing Test Results.
- Section C.3: Influence of Dialogue Length on Turing Test Performance.

### C.1   TURING TEST PLATFORM

***Pre-test phase***. Prior to the evaluation, participants provide basic demographic information as shown in Figure 9a, including age, gender, education, and familiarity with AI, which may influence their judgments. They can also select between Chinese and English dialogues, allowing them to make judgment using their preferred language and thereby improving the reliability of the results. ***Testing phase***. Each round of evaluation contains 5 dialogues to be judged. After completing a round, participants may either proceed to the next round or pause. ***Post-test phase***. All incomplete submissions

are discarded to ensure data integrity. The remaining responses are then aggregated and analyzed in conjunction with the demographic information collected during the pre-test phase. To boost engagement, participants receive points based on the accuracy of their judgments, and a public leaderboard ranks all players based on their performance as shown in Figure 9c. This analysis enables us to identify potential influences of user characteristics on evaluation outcomes. The homepage and main interface of the platform are illustrated in Figure 9b and Figure 3, respectively.

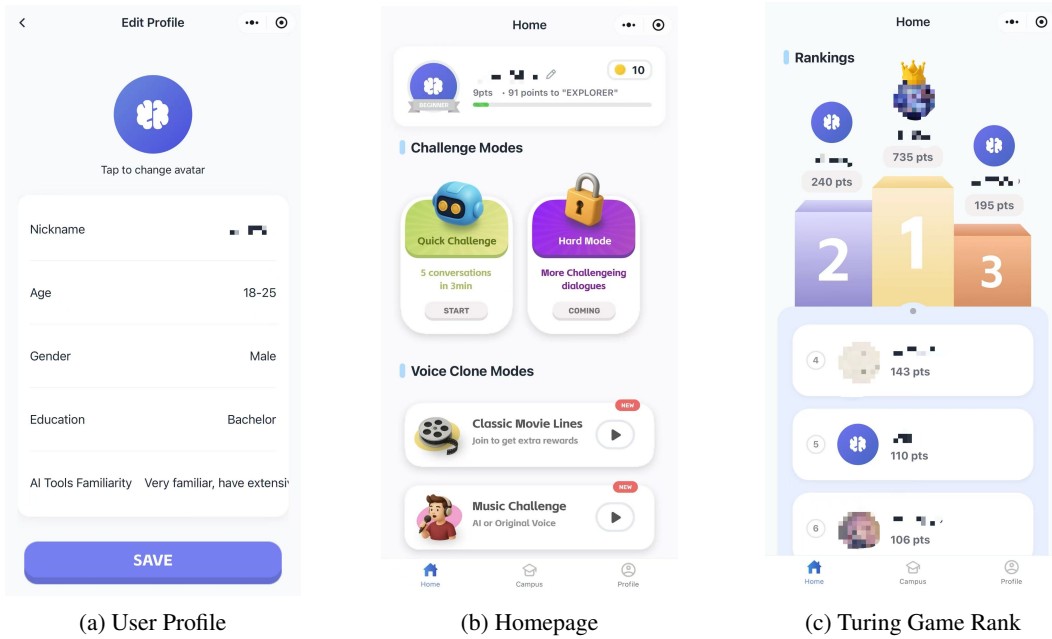

(a) User Profile          (b) Homepage          (c) Turing Game Rank

Figure 9: The Turing test platform.

## C.2 SUPPLEMENTARY TURING TEST RESULTS

Table 9 shows the exact Turing test success rates of S2S systems, measured as the proportion of responses judged as human. Higher values indicate greater human-likenes.

Table 9: Success rate of S2S systems (%).

| Model | GPT-4o | Claude-Sonnet 4 | Qwen3 | Gemini-2.5 pro | Kimi-K1.5 | ChatGLM-1.5 |
|---|---|---|---|---|---|---|
| **English** | 25.9 | 22.9 | 6.7 | 19.0 | 30.8 | 11.8 |
| **Chinese** | 23.0 | 0.0 | 16.4 | 13.3 | 11.0 | 9.6 |
| **Model** | **HunyuanTurboS** | **Doubao-Pro 1.5** | **iFLYTEK-Spark** | **Spark-TTS** | **Nari-TTS** | **Human Speaker** |
| **English** | 20.0 | 21.9 | 0.0 | 25.6 | 37.8 | 86.7 |
| **Chinese** | 20.9 | 21.9 | 14.0 | 36.6 | 0.0 | 70.0 |

Figure 10 presents the success rates of S2S systems by levels.

## C.3 INFLUENCE OF DIALOGUE LENGTH ON TURING TEST PERFORMANCE

We divided the Turing test results by dialogue length and calculated the classification accuracy for different dialogue types: human-human dialogues (H-H), human-machine dialogues (H-M), and pseudo-human dialogues (PH). Table 10 summarizes the accuracy results across different duration ranges.

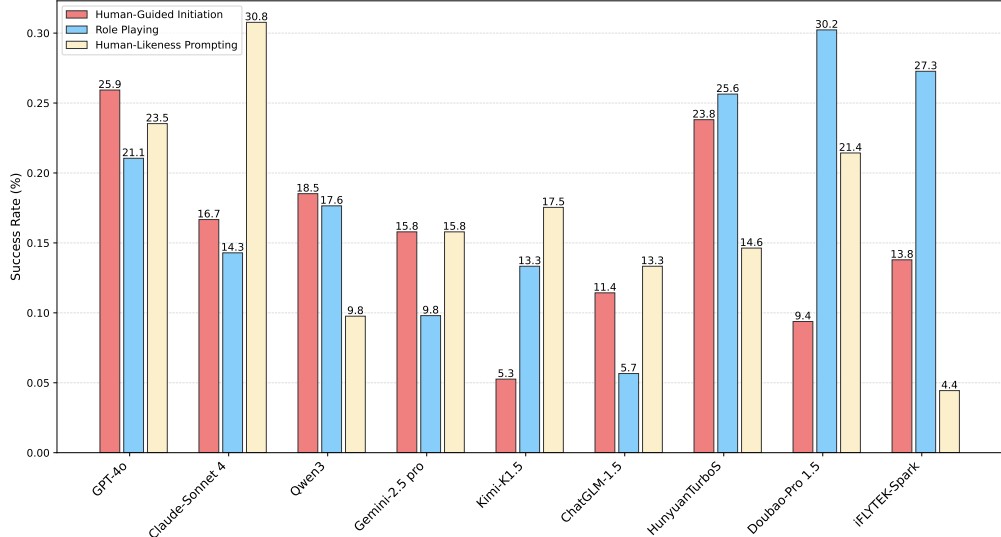

Figure 10: Success rate of S2S systems by different interaction strategies.

Table 10: Accuracy by Duration Interval for H-H, H-M, and PH.

| Duration | H-H (acc/count) | H-M (acc/count) | PH (acc/count) |
|----------|-----------------|-----------------|----------------|
| [20,25)  | 0.4000 / 5      | – / 0           | 0.6624 / 157   |
| [25,30)  | 0.7800 / 50     | 0.7742 / 31     | 0.6654 / 257   |
| [30,35)  | 0.6513 / 152    | 0.8333 / 126    | 0.6337 / 243   |
| [35,40)  | 0.7033 / 246    | 0.8642 / 162    | 0.6087 / 92    |
| [40,45)  | 0.6839 / 174    | 0.8498 / 273    | 0.6200 / 50    |
| [45,50)  | 0.7179 / 78     | 0.8564 / 195    | 0.6333 / 60    |
| [50,55)  | 0.8421 / 76     | 0.7737 / 137    | 0.5349 / 43    |
| [55,60)  | 0.7234 / 141    | 0.7907 / 43     | – / 0          |

As shown in Table 11, we performed Cochran–Armitage Trend Tests to examine the potential linear relationship between dialogue length and accuracy and found no significant trend for any individual dialogue type. This suggests that dialogue length alone does not significantly influence the likelihood of passing the Turing test.

Table 11: Statistical Test Results Across Dialogue Types.

| Dialogue Type | Z Statistic | p-value | Significant Trend? |
|---------------|-------------|---------|--------------------|
| H-H           | 1.6604      | 0.09683 | ✗                  |
| H-M           | -1.0106     | 0.31220 | ✗                  |
| PH            | -1.6018     | 0.10919 | ✗                  |

# D  FINE-GRAINED HUMAN-LIKENESS DIMENSIONS

The section is organized into the following sections:

- Section D.1: The Taxonomy for Fine-Grained Human-Likeness Diagnosis.

- Section D.2: Annotation Process.

- Section D.3: Annotation Quality Assurance.

## D.1 The Taxonomy for Fine-Grained Human-Likeness Diagnosis

We organize the evaluation dimensions into five categories: *I. Semantic Features and Pragmatic Habits; II. Non-Physiological Paralinguistic Features; III. Physiological Paralinguistic Features; IV. Mechanical Persona; V. Emotional Expression*. Notably, annotators are instructed to use these dimension descriptions to rate the human-likeness of each conversational response on a five-point scale: 1 indicates strongly machine-like behavior, 5 indicates strongly human-like behavior, and 3 denotes no clear human- or machine-like leaning, or no enough evidence to judge.

First, five speech domain experts employ a prompt-driven, heuristic querying process with GPT-4o to generate an initial set of concepts that differentiate human and machine responses. This method ensures that the generated concepts are both grounded in expert knowledge and supported by the model's comprehensive language understanding. The set is then refined iteratively, with expert feedback and relevant social science literature retrieved by GPT-4o, ensuring that only the most representative and discriminative concepts are retained. The resulting dimensions are summarized in Table 12. This refinement process adds scientific rigor by aligning the selected concepts with established theories in the field, enhancing their validity. The final outcome is a set of five categories, encompassing 18 fine-grained dimensions, which are both comprehensive and precise. These dimensions were subsequently used to annotate dialogue training data through a crowdsourcing model. The ultimately trained model achieved a significant improvement in human-machine dialogue recognition. This also demonstrates the reasonableness and reliability of these dimensions.

Table 12: Fine-grained human-likeness evaluation taxonomy .

| Dimension | Description |
|---|---|
| *Memory Consistency (I)* | Machine-like: Forgetting key information and unable to realize errors; Human-like: Consistent memory in short contexts or asks for clarification when misunderstanding occurs(Toneva et al., 2019). |
| *Logical Coherence (I)* | Machine-like: Abrupt logical transitions or self-contradictions; Human-like: Natural and coherent reasoning(Bottazzi Grifoni & Ferrario, 2025). |
| *Pronunciation Accuracy (I)* | Machine-like: Mispronunciation (including heteronyms); Human-like: Correct pronunciation, with proper usage of heteronyms(Zhang, 2021). |
| *Code-switching (I)* | Machine-like: Unreasonable multilingual mix; Human-like: The mix of languages is context-dependent, and the switching is smooth(Zhang, 2019). |
| *Linguistic Imprecision (I)* | Machine-like: Responses are precise and affirmative; Human-like: Uses vague expressions(Piantadosi et al., 2012) like "probably", and self-corrections(Nakatani & Hirschberg, 1993). |
| *Use of Fillers (I)* | Machine-like: Rare use of fillers or unnatural usage; Human-like: Frequently uses (e.g., "um", "like") while thinking(Székely et al., 2019). |
| *Metaphor & Implied Meaning (I)* | Machine-like: Direct, lacking semantic diversity, only capable of surface-level interpretation; Human-like: Uses metaphor and euphemism to convey implied meanings(Vulchanova & Vulchanov, 2018). |
| *Rhythm (II)* | Machine-like: No pauses or mechanical pauses; Human-like: Speaking rate varies with semantic coherence, with occasional hesitations(Hwang et al., 2023). |
| *Intonation (II)* | Machine-like: Unnatural or flat intonation; Human-like: Natural pitch rise or fall(Warren et al., 2025). |
| *Stress (II)* | Machine-like: No emphasis on words or abnormal emphasis placement; Human-like: Consciously emphasizes key words(Prieto & Roseano, 2018). |
| *Auxiliary Vocalizations (II)* | Machine-like: Contextually incorrect or mechanical auxiliary sounds; Human-like: Produces appropriate non-verbal sounds to express emotion(Anikin et al., 2023). |
| *Micro-physiological Noise (III)* | Machine-like: Speech is overly clean or emits unnatural sound; Human-like: Humans produces breathing sounds, saliva sounds, etc(Fukuda et al., 2018). |

| *Pronunciation Instability (III)* | Machine-like: Pronunciation is overly clear; Human-like: Some irregularities in pronunciation (e.g., tremolo, slurred speech, nasal sounds)(Teixeira et al., 2013). |
|---|---|
| *Accent (III)* | Machine-like: Stiff and unnatural accent; Human-like: Natural regional accent or vocal traits(Onda et al., 2025). |
| *Sycophant Behavior (IV)* | Machine-like: Excessively agrees, thanks, and apologizes; Human-like: Judges whether to agree based on context(Fanous et al., 2025). |
| *Written-style Expression (IV)* | Machine-like: Responses are well-structured and formal. frequent listing; Human-like: Conversational, flexible, and varied expression(Doyle et al., 2019). |
| *Textual Sentiment (V)* | Machine-like: Emotion conveyed in text may appear mismatched with natural human sentiment. Human-like: Emotion in text feels authentic and resonates naturally with human emotional expression.(Wang et al., 2025a). |
| *Acoustic Emotion (V)* | Machine-like: Prosody or tone may sound inconsistent with the intended emotion expression of the text. Human-like: Vocal delivery conveys context-appropriate emotional cues that align with the text(Voorveld et al., 2025). |

## D.2 ANNOTATION PROCESS

We recruited 36 annotators, all of whom are graduate students (master's or Ph.D.) specializing in artificial intelligence, with bilingual fluency in both English and Chinese. Prior to the formal annotation, each annotator received comprehensive annotation guidelines (see Figure 11) and completed a calibration phase consisting of multiple trial batches (5 samples per batch, approximately 20–30 minutes each) to ensure consistent understanding of the evaluation criteria. Annotators were compensated at a rate of 30 units/hour (local currency), with a total cost equivalent to approximately 5,250 units.

Annotators are instructed to use these dimension descriptions to rate the human-likeness of each conversational response on a five-point scale: 1 indicates strongly machine-like behavior, 5 indicates strongly human-like behavior, and 3 denotes no clear human- or machine-like leaning, or no enough evidence to judge. In line with the setup of Turing Test, they only evaluated the responder's performance in each dialogue.

To ensure the reliability of annotations, we implemented the following:

- We created a questionnaire webpage where the crowdsourced annotators could access. Screenshots of the web are provided as figure11 and figure12. Annotators' submissions are stored in our private Hugging Face dataset. Each submission contains 5 dialogues. For each dialogue, 18 ratings based on the 18 dimensions are associated with it. After grading each dialogue, the annotators also needed to indicate their judgment of the identity of the responder (final choice).

- We provided reference descriptions as mentioned in the previous section for each dimension.

- The annotators were unaware of the human-machine identity of the responder, and has never heard the dialogues before.

- Before scoring, annotators must read the detailed guidelines, and we also provided training and clarification for them.

**Guideline for Annotators**

- The score reflects the degree of human-likeness of the response in a given dimension.

- Even if you are confident about the identity of the responder, you are required to independently evaluate the degree of human-likeness for different dimensions.

- A score of 3 indicates uncertainty about whether the responder is more human-like or machine-like. It also indicates that the dimension was not reflected in the dialogue. The underlying meaning of 3 is that this score has no contribution to the final choice.

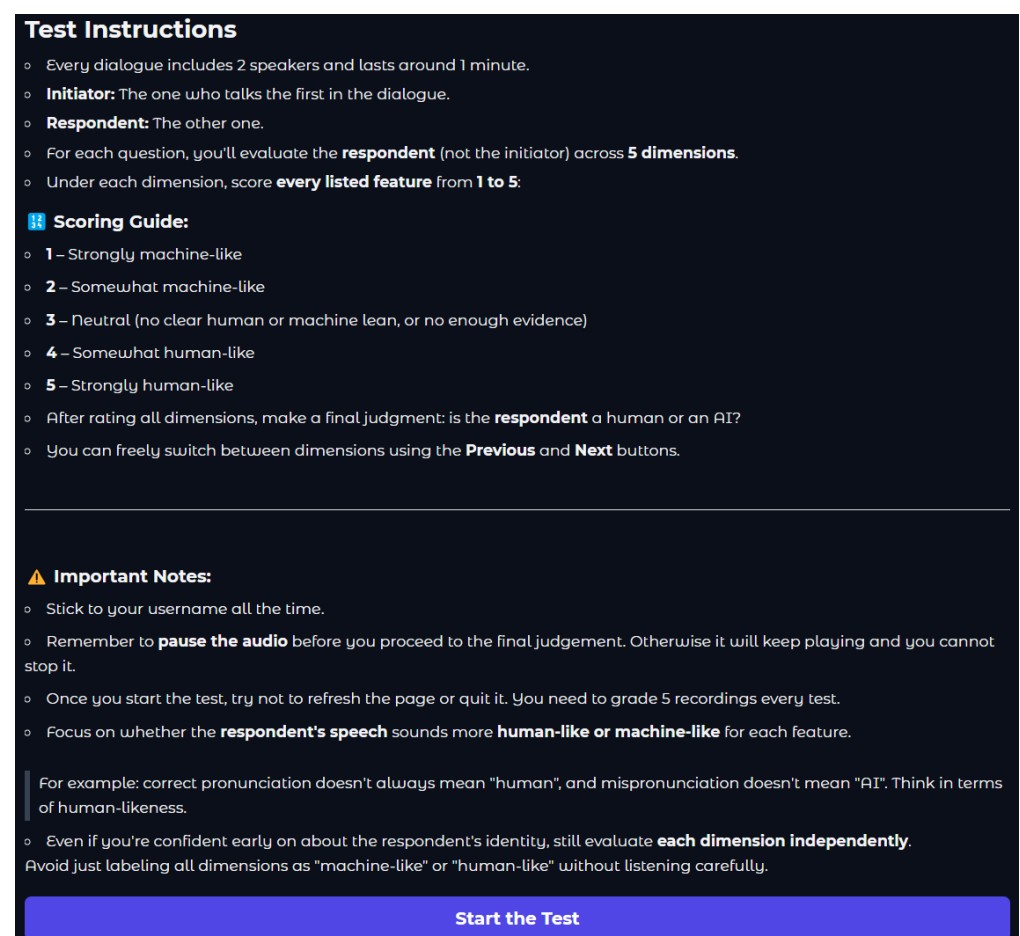

Figure 11: Annotator guideline page.

## D.3    ANNOTATION QUALITY ASSURANCE

For quality control, we invited three experts specializing in human-computer interaction to conduct cross-validation on all submitted content. Each expert was provided with the true labels indicating whether the dialogue was generated by a human or a machine. Only annotations unanimously approved by all three experts were included, while those with any disagreement underwent expert discussion for revision. A total of 29.44% of the labels were revised, with an average adjustment of 1.99 points (49.76% of the score range), demonstrating the effectiveness of expert review in mitigating noise in the raw annotations. Table 13 presents the three dimensions with the highest change ratio, along with overall results across all 18 dimensions.

Table 13: Expert Revision Impact on Label Adjustments

| Dimension | Change Ratio | RMSE | RMSE Ratio |
|---|---|---|---|
| Pronunciation Accuracy | 0.3596 | 2.1085 | 0.5271 |
| Textual Sentiment | 0.3472 | 1.9579 | 0.4895 |
| Linguistic Imprecision | 0.3273 | 2.1230 | 0.5308 |
| Overall | 0.2944 | 1.9903 | 0.4976 |

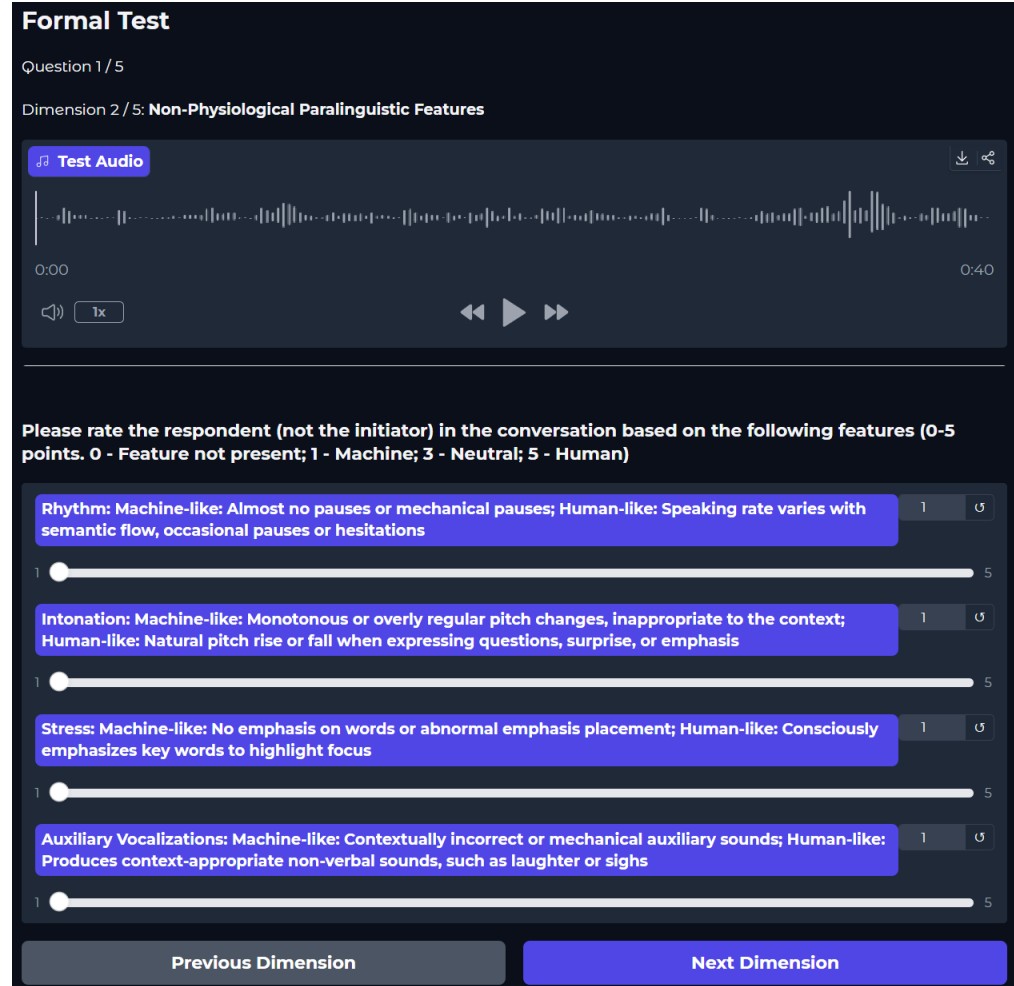

Figure 12: Annotation page example.

To further validate annotation reliability, we trained models on data before and after expert correction. As shown in Table 14. Expert-refined labels lead to substantial improvements in both in-distribution and OOD generalization, confirming the quality of our final annotation set.

Table 14: Binary Classification Accuracy (Before/After Expert Modification)

| Data | Overall (Inner) | CosyVoice2 | Fisher | MultiDialog | Overall (External) |
|------|-----------------|------------|--------|-------------|--------------------|
| Original | 0.8791 | 0.9375 | 0.6250 | 0.9062 | 0.8229 |
| Modified | **0.9605** | 0.9844 | 0.9844 | 0.9531 | **0.9740** |

# E  EXPERIMENT DETAILS OF AI JUDGER

The section is organized into the following sections:

- Section E.1: Prompt Templates for AI Judges.
- Section E.2: Training Setup.
- Section E.3: Embedding Readout Selection.
- Section E.4: Model Ablation.

- Section E.5: Hyperparameter Tuning.
- Section E.6: Fine-Grained Human-Likeness Scoring Accuracy.
- Section E.7: Contribution Analysis by Case Study.

## E.1 Prompt Templates for AI Judges

The following Figure 13 shows the prompt used for AI judges.

> You are an expert in speech analysis and speaker identity recognition. Your core functions include but are not limited to: Automatic Speech Recognition, acoustic feature extraction, speaker diarization, and analysis of Text-to-Speech technologies. You are designed to perform objective, evidence-based reasoning. Analyze the dialogue in this audio file and strictly adhere to the following steps:
> - Speaker Diarization: Identify and differentiate the speech segments of Speaker A (the first person to speak) and Speaker B (the respondent).
> - AI-Generated Voice Detection: Based on an analysis of vocal characteristics, determine whether the voice of Speaker B (the respondent) is AI-generated.
> - Verdict:
>     - If Speaker B's voice is determined to be AI-generated, output:
>       `final_choice="True"`.
>     - Otherwise, output: `final_choice="False"`.

Figure 13: Prompt templates for AI judges.

## E.2 Training Setup

All experiments are conducted on our constructed dataset. Specifically, we use 831 samples ($\approx$11h) for training and 208 samples ($\approx$2h) for validation, obtained from the H-H and H-M subsets with a 1:1 ratio. The test set consists of the remaining Human-Human(H-H) and Human-Machine(H-M) samples together with TTS data, forming 430 samples ($\approx$5h) with a balanced 1:1:1 distribution. For modeling, we adopt **Qwen2.5-Omni-7B** as the backbone of our turing judge and further evaluate its LoRA fine-tuned variant. During hidden state extraction, we fix $\text{random\_sample} = False$ to ensure consistency, and apply standard normalization to the hidden representations. For both modules, we adopt Adam as optimizer. The complete experiments, covering feature extraction, inference evaluation of multimodal large models, and model training, are carried out on a computing cluster with 8×A40 GPUs (48 GB memory per GPU).

## E.3 Embedding Readout Selection

**Readout Design.** In Qwen2.5-Omni, only the first step exposes hidden states for the *complete* input sequence; at subsequent steps, each layer outputs a hidden state only for the *newly generated* token. Under this constraint, we design three readout candidates:: (i) **First-step mean pooling**: a simple average over step-1 token-level states (a length-agnostic baseline); (ii) **Last-token representation**: the hidden state of the most recent token as a compact, compression-style summary; and (iii) **Fused pooling**: a learnable weighted fusion of {*first_hidden_mean*, *last_hidden*} into a single embedding. This two-source hidden representation lets the model adaptively fuse the globally contextual, acoustics-aware signal in *first_hidden_mean* with the high-level semantics distilled in *last_hidden*.

**Ablation study.** To identify which sequence embedding best supports our downstream objectives, we conduct an ablation under a unified hyperparameter regime (Table 15). We evaluate the three

readout strategies under fixed protocols so that any performance differences can be attributed solely to the readout. Each alternative is paired with the same ODL-Linear head and trained end to end to convergence. To assess stability, all evaluations are conducted

Table 15: Tuning parameters.

| Prompt | Scale | Batch Size | Learning Rate | Dropout |
|---|---|---|---|---|
| Understanding | 4 | 64 | $1 \times 10^{-4}$ for ODL $1 \times 10^{-3}$ for Linear | 0.1 |

five times with different random seeds; we report the human–machine classification accuracy as the mean ± standard error (s.e.m.) over the five runs.

Table 16 shows the overall performances corresponding to three readouts. Fused pooling attains the highest overall score (0.9112), outperforming mean pooling (0.8879) and last-token representation (0.8032). On the *Pseudo Human* dataset—which is strictly out-of-distribution—fused pooling reaches 0.8167, while other baselines remain at 0.7805 and 0.7917. This gap indicates improved robustness to distributional shift. Consistent gains across Human-Human and Human-Machine settings further demonstrate that the fusion of abstract semantic and speech sequential information enhances discriminative power beyond what either source provides independently.

Table 16: Ablation experiment results (mean $\pm$ s.e.m. over 5 runs).

| Data Type | First-step Mean Pooling | Last-token Representation | Fused Pooling |
|---|---|---|---|
| Human-Human↑ | 0.9409(±0.0017) | 0.7380(±0.0047) | **0.9493(±0.0014)** |
| Human-Machine↑ | **0.9430(±0.0051)** | 0.8791(±0.0028) | 0.9306(±0.0017) |
| Pseudo Human ↑ | 0.7805(±0.0100) | 0.7917(±0.0022) | **0.8167(±0.0061)** |
| Overall ↑ | 0.8879(±0.0044) | 0.8032(±0.0012) | **0.9112(±0.0020)** |

Overall, these findings show that the choice of readout materially impacts downstream performance. Fused pooling provides consistent improvements across all settings, including out-of-distribution evaluation, and therefore constitutes a reliable default for sequence-level embedding utilization.

### E.4 MODEL ABLATION

To validate the effectiveness of ODL, we conducted an ablation where we removed the ODL and replaced it with a standard linear layer and negative log-likelihood loss, treating the human-likeness scores as independent categories. This baseline corresponds to a non-ordinal but still interpretable classifier. This clarifies that ODL is used as an appropriate modeling choice for ordinal labels, and that our ablation demonstrates its empirical value.

Table 17: Binary classification accuracy across module ablation

| Projection Module | Human-Human | Human-Machine | Pseudo Human | Overall |
|---|---|---|---|---|
| **Ordinal Discretization Layer** | 0.9507 | 0.9722 | 0.9306 | **0.9605** |
| **Linear Layer** | 0.8718 | 0.9875 | 0.9097 | **0.9233** |

### E.5 HYPERPARAMETER TUNING

**Grid Search.** To further optimize model's performance, we tune hyperparameters for ODL and FL independently using grid search.

As summarized in Table 18, the ODL space comprises $3 \times 1000 \times 4 \times 4 \times 5 = 240,000$ configurations, while the FL space contains $4 \times 4 = 16$. The joint search space therefore consists of 3.84M combinations. To reduce computational cost, we uniformly sampled 7500 ODL configurations and paired each with all 16 FL settings, yielding 120,000 trials in total. Each trial requires $\sim$5 minutes on a single GPU, corresponding to $\sim 10,000$ GPU-hours overall.

**Tuning Criterion.** To select optimal hyperparameters, we adopt accuracy as the primary objective for grid search, which reflects the downstream classification goal of human–machine discrimina-

Table 18: Hyperparameter search space.

| Module | Prompt | Scale | Batch Size | Learning Rate | Dropout |
|---|---|---|---|---|---|
| Ordinal Discretization Layer | Understanding Transcribe Classify | $1:0.01:10$ | 16 32 64 128 | 1e-2 1e-3 1e-4 1e-5 | 0.1 0.2 0.3 0.4 0.5 |
| Linear Layer | – | – | 32 64 128 256 | 1e-2 1e-3 1e-4 1e-5 | – |

tion. The tuning results are summarized in Table 19, and the selected configuration is used used throughout all experiments.

Table 19: Tuning results.

| Module | Prompt | Scale | Batch Size | Learning Rate | Dropout |
|---|---|---|---|---|---|
| Ordinal Discretization Layer | Understanding | 2.1 | 64 | 1e-5 | 0.3 |
| Linear Layer | – | – | 128 | 1e-3 | – |

**Sensitivity Analysis.** As a complementary experiment to our main hyperparameter tuning, we performed a 1000-run randomized hyperparameter search, sampling key training parameters for ODL (learning rate, batch size, scale, dropout) and FL (learning rate, batch size). Each configuration was trained end-to-end using the same evaluation protocol, ensuring reliability through full parallelization. The results for the hyperparameter sensitivity analysis (accuracy) are presented in the Table 20.

Table 20: Hyperparameter Sensitivity Evaluation Metrics

| Hyperparameter | Values | Acc (ODL) | Acc (FL) | MSE (ODL) | MSE (FL) |
|---|---|---|---|---|---|
| lr (ODL) | {1e-05, 1e-04, 1e-03, 1e-02} | 0.6020(±0.0435) | 0.8642 (±0.0071) | 0.002174 | 0.000051 |
| batch_size (ODL) | {32, 64, 128, 256} | 0.6105 (±0.0065) | 0.8601(±0.0149) | 0.000048 | 0.000222 |
| scale | {1, 1.05, …, 5} | 0.6293(±0.0090) | 0.9254(±0.0283) | 0.000091 | 0.000802 |
| dropout | {0.1, 0.2, 0.3, 0.4, 0.5} | 0.6103 (±0.0050) | 0.8617 (±0.0103) | 0.000029 | 0.000105 |
| lr (FL) | {1e-05, 1e-04, 1e-03, 1e-02} | 0.6109(±0.0031) | 0.8584(±0.0696) | 0.000011 | 0.004838 |
| batch_size (FL) | {16, 32, 64, 128} | 0.6107(±0.0034) | 0.8650(±0.0193) | 0.000013 | 0.000371 |

Analyzing the results, we identify several key findings:

- Learning rate proved to be a critical factor for both ODL and FL, consistent with findings from other work. Extremes caused underfitting or instability, emphasizing the need for precise tuning.
- Scale had minimal impact on ODL accuracy, suggesting ODL's adaptability, but slightly affected FL due to scale-induced changes in logits cut-points.
- Batch size influenced FL performance, with larger batches stabilizing training but potentially slowing convergence or causing overfitting.
- Dropout and ODL batch size showed minimal effects

Overall, the 1000-run analysis shows that our method is generally robust, with learning rate being the most sensitive parameter, while other hyperparameters produce only modest effects.

### E.6 FINE-GRAINED HUMAN-LIKENESS SCORING ACCURACY

**Accuracy Analysis.** Since the 1–5 scores reflect perceived human-likeness, we report not only the *exact* accuracy that measures full agreement with human judgments, but also a *grouped* accuracy that consolidates scores into three categories (1–2, 3, and 4–5) to better reflect alignment with human perception. In addition, we include accuracy within a tolerance of ±1 to capture near-agreement with human ratings.

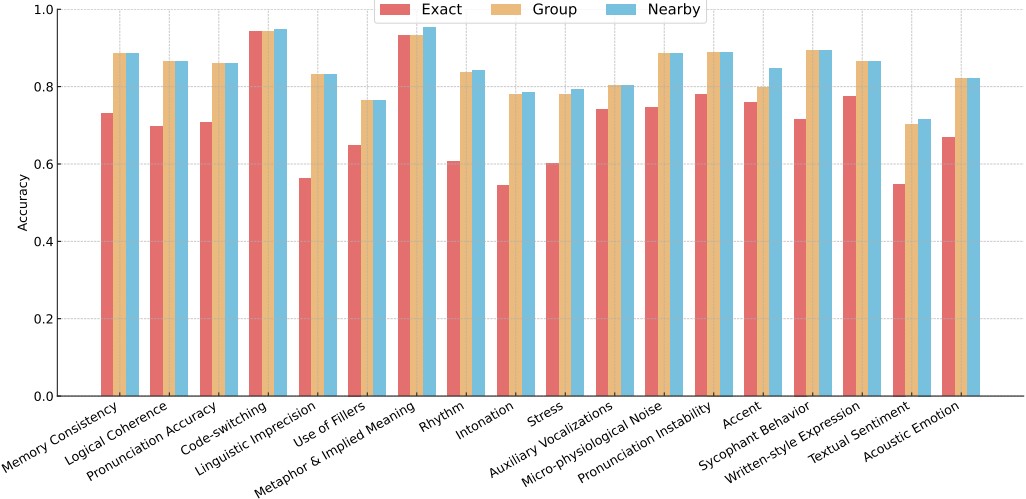

Figure 14: Fine-grained scoring accuracy.

As shown in Figure 14, the Ordinal Discretization Layer consistently exceeds 50% *exact* accuracy across all evaluation dimensions, often reaching 70%. When consolidating scores into three bins or allowing a tolerance of ±1, accuracies in most dimensions approach or surpass 80%. With more detailed accuracies provided in Table 21, these results indicate that the model captures the correct ordinal direction in fine-grained judgments and aligns closely with human perceptions, yielding interpretable evidence for downstream human–machine classification. Moreover, our training framework not only substantially enhances binary classification accuracy but also systematically aligns the model with human evaluation dimensions, enabling it to learn human-like judgment patterns.

Table 21: Detaied accuracies.

| Metrics\Dim | MC | LC | PA | CS | LI | UF | MM | RT | IT |
|---|---|---|---|---|---|---|---|---|---|
| ACC↑ | 0.7308 | 0.6971 | 0.7067 | 0.9423 | 0.5625 | 0.6490 | 0.9327 | 0.6058 | 0.5433 |
| ACC (Group)↑ | 0.8846 | 0.8654 | 0.8606 | 0.9423 | 0.8317 | 0.7644 | 0.9327 | 0.8365 | 0.7788 |
| ACC (±1)↑ | 0.8846 | 0.8654 | 0.8606 | 0.9471 | 0.8317 | 0.7644 | 0.9519 | 0.8413 | 0.7837 |

| Metrics\Dim | ST | AV | MN | PI | AC | SB | WE | TS | AE |
|---|---|---|---|---|---|---|---|---|---|
| ACC↑ | 0.6010 | 0.7404 | 0.7452 | 0.7788 | 0.7596 | 0.7163 | 0.7740 | 0.5481 | 0.6683 |
| ACC (Group)↑ | 0.7788 | 0.8029 | 0.8846 | 0.8894 | 0.7981 | 0.8942 | 0.8654 | 0.7019 | 0.8221 |
| ACC (±1)↑ | 0.7933 | 0.8029 | 0.8846 | 0.8894 | 0.8462 | 0.8942 | 0.8654 | 0.7163 | 0.8221 |

**Out-of-domain Evaluation** To further evaluate the model's generalization for the five-degree rating, we invited human experts to annotate the OOD samples on multiple dimensions and report three accuracy metrics, where Exact is the percentage of predictions that exactly match the expert score, Group is the percentage that fall into the same human–machine identity group (1–2 machine-like, 3 unclear, 4–5 human-like), and Nearby is the percentage that differ from the expert score by at most ±1. The results are shown in the Table 22, indicating that our model maintains strong generalization ability in fine-grained scoring.

Table 22: Overall Fine-grained Scores Accuracy

| Dataset | Exact | Group | Nearby |
|---|---|---|---|
| Ours | 0.7056 | 0.8408 | 0.8470 |
| CosyVoice2 | 0.6450 | 0.7569 | 0.8030 |
| Fisher | 0.6476 | 0.7396 | 0.7752 |
| MultiDialog | 0.6562 | 0.7561 | 0.7847 |

## E.7 CONTRIBUTION ANALYSIS BY CASE STUDY

**Case Study.** To probe the interpretability of the model's human–machine discrimination, we conduct case studies spanning two diagnostic regimes: (i) *machine-class true positive* (instances correctly predicted as machine) and (ii) *machine-class false negative* (machine instances incorrectly predicted as human). This design reflects and operationalizes the principles of the inverted Turing test, establishing continuity between our analytical setting and evaluation framework.

For each instance, we first calculate each contribution $c_k$ on machine-side by producing standardized ODL logits (standardized with respect to the training-set distribution) together with corresponding trained linear weight. Then, we rank top 8 features by $|c_k|$ to identify the most influential factors. By construction, $c_k > 0$ (machine-like scoring) increases evidence for the machine class, whereas $c_k < 0$ (human-like scoring) reduces it.

As shown in Figure 15, most fine-grained scores align with their final contributions to human–machine classification. In Figure 15a, despite a strong human-like cue (e.g., a negative contribution from *Pronunciation Accuracy*), the model aggregates multiple machine-oriented signals, such as *Sycophant Behavior* and *Pronunciation Instability*, yielding a high-confidence correct decision. By contrast, in Figure 15b, high-score dimensions (e.g., *Memory Consistency*, *Pronunciation Accuracy*) contribute salient human-like evidence that shifts a machine sample into the human region; the available machine-like cues are insufficient to overturn the outcome due to a small effective margin, leading the system to accept the machine response as human in the sense of an inverted Turing test.

Case evidence shows that S2S outputs perform strongly on dimensions such as *Memory Consistency* and *Logical Coherence*, leading annotated scores to concentrate in the 4–5 range; nevertheless, the associated logits remain informative within this high-score regime. When the model maps inputs to human-like scores, these dimensions place samples within higher-valued latent intervals along a continuous scoring axis. This induces within-bin margins: sample-wise logit variability driven by subtle linguistic or acoustic cues. In downstream binary classification, such variability produces margin-dependent contributions: near-cutpoint (low-margin) instances can exert *negative* influence, whereas far-beyond-cutpoint (high-margin) instances provide *strong positive* evidence. Thus, even under apparent rating saturation, logits retain fine-grained discriminative power via their ordinal positions and margins.

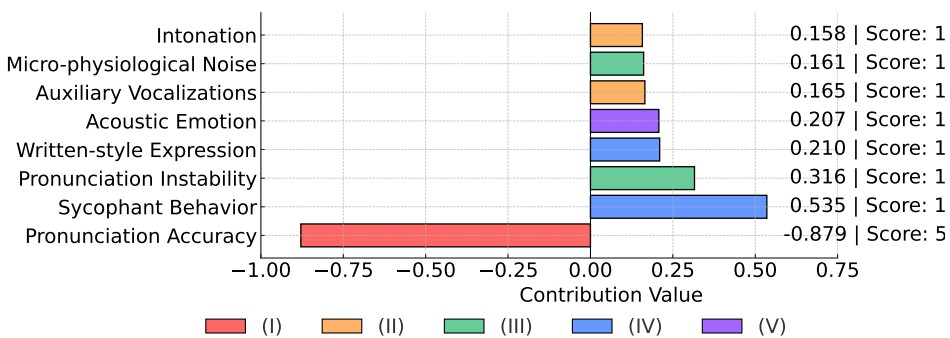

(a) Machine-class true positive

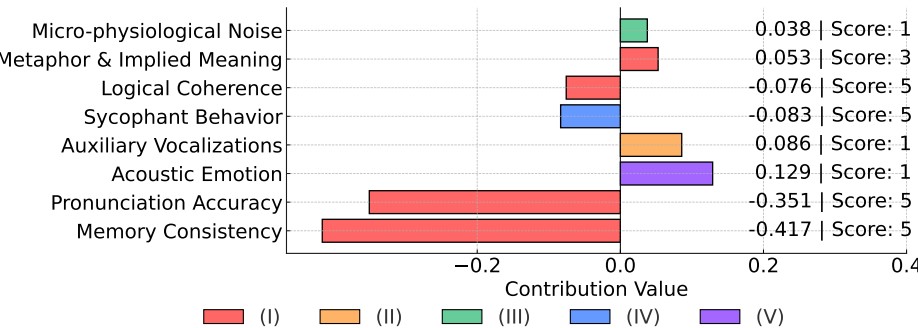

(b) Machine-class false negative

Figure 15: Case studies

