# OpenReview forum: "Human or Machine? A Preliminary Turing Test for Speech-to-Speech Interaction"
_ICLR.cc/2026/Conference — ICLR 2026 Poster_

### Official Review · Reviewer_3UQQ · 2025-10-30

**Soundness:** 2
**Presentation:** 2
**Contribution:** 2
**Rating:** 2
**Confidence:** 4

**Summary:**

This paper presents a study of testing if current speech-to-speech (S2S) models, e.g., GPT-4o's advanced voice mode, can pass a Turning
test of conducting human-like conversations. Authors firstly constructed S2S dialog datasets between human-model conversations, as
well as synthesizing speeches with TTS models. After that, approximately 3K human judgements were carried out, measuring the human-likeness of current S2S models. To measure the human-likeness, authors defined 18 dimensions such as memory consistency, use of fillers etc.

Authors show that humans can easily identify human-model dialogs, such that current S2S models cannot pass the Turning test. Authors also show that off-the-shelf AI models cannot serve as a judge for the test; finetuning the off-the-shelf AI on the authors' created dialog datasets enable AI to better judge human-model dialogs.

**Strengths:**

- This is the first evaluation of S2S models from the Turing test prospective, which is new and interesting.
- Evaluation details are well presented with clear takeaway messages, and the paper is easy to follow.

**Weaknesses:**

- It seems the conclusion, i.e., current S2S models cannot pass Turning test, is very much expected.  E.g., VoiceBench[1] has shown that current S2S models still largely lag behind their text counterparts. Then the main contribution seems to be the "Turing Test" framing. I think authors should more clearly demonstrate how this paradigm can provide more insights than existing S2S benchmarks like VoiceBench.

- The generalization ability of the judge model finetuned on authors' created datasets is unknown. It is very much expected that the in-domain (e.g,  based on the 18 evaluation dims) finetuned judge model performs the best on judging the dialogs. Authors should test the correlation of  the judge model with humans, when being applied to some other real-world OOD dialogs to ensure the generalization ability, beyond only testing on the pseudo-human datasets.

- Some claims seem contradicting (cf Q2).

[1] VoiceBench: Benchmarking LLM-Based Voice Assistants

**Questions:**

- Line140: The datasets seem contain both English and Chinese. Are the 28 participants from 10 countries all speak both two languages?

- Line278-279 describe that current S2S are limited by aspects like topic understanding etc. While Line332 claims that S2S models have largely solved the foundational challenges of understanding and generating clear and coherent dialogue turns.  Wouldn't these claims contradicting? How do you define semantic and paralinguistic tasks?

---

> ### Author Response · Authors · 2025-11-19
> **Replies to Reviewer 3UQQ**
>
> We sincerely thank the reviewer for recognizing the strengths of our work, including that it is *“the first evaluation of S2S models from the Turing test perspective, which is new and interesting,”* and that *“the evaluation details are well presented with clear takeaway messages and the paper is easy to follow.”* Below, we address each concern in detail.
>
> **Weakness 1: Comparison with Existing Speech Benchmarks**
>
> We conducted a detailed comparison between our work and two representative benchmarks, VoiceBench [1] and MMAU-Pro [2]. As summarized in the table below, our work differs fundamentally in evaluation goal and evaluated modality.
>
> | Aspect | VoiceBench | MMAU-Pro | (S2S) Turing Test (**Ours**) |
> | --- | --- | --- | --- |
> | Goal | Evaluating speech understanding in LLM-based voice assistants | Evaluating holistic audio understanding of multimodal AI models across speech, music, and sound. | Evaluating Human-likeness of Speech-to-Speech Systems |
> | Input Modality | Speech or Text | Speech and Text | Speech |
> | Outout Modality Evaluated On | Text | Text | Speech |
> | Dialogue Turns | Single-turn | Multi-turn | Multi-turn |
> | The Smarter the Better? | Yes — higher intelligence implies better performance | Yes — higher intelligence implies better performance | No — being “too smart” does not necessarily make a model more likely to pass the Turing Test. |
>
> To further examine whether “being smarter” makes a model more human-like, we selected S2S systems that appear in both MMAU-Pro and our study, and compared their reasoning accuracy on MMAU-Pro with their Turing Test pass rates. The results are summarized below.
>
>
> | Model          | Reasoning Accuracy (MMAU-Pro) | Turing Test Pass Rate |
> |---------------------|----------------------------------------|----------------------------|
> | Kimi-K1.5       | 46.6                                   | 12.7                     |
> | Qwen3          | 52.2                                   | 15.1                     |
> | GPT-4o        | 52.5                                   | 23.0                     |
> | Gemini-2.5-Pro  | 59.2                                   | 13.7                     |
>
> Pearson correlation (Reasoning Accuracy ↔ Turing Test Pass Rate): 0.0456
>
> This indicates that reasoning ability is nearly uncorrelated with human-likeness in current S2S systems, revealing a disconnect between traditional intelligence benchmarks and the human-likeness required for speech interaction.
>
> [1] Chen, Yiming, et al. Voicebench: Benchmarking llm-based voice assistants. arXiv preprint arXiv:2410.17196 (2024).
>
> [2] Kumar, Sonal, et al. Mmau-pro: A challenging and comprehensive benchmark for holistic evaluation of audio general intelligence. arXiv preprint arXiv:2508.13992 (2025).
>
> #### **Weakness 2: Generalization ability of the judge model**
>
> We evaluated the generalization of our judge model using an OOD dataset, with 64 dialogues sampled from CosyVoice2 [3], Fisher [4], and MultiDialog [5], respectively. CosyVoice2 denotes machine-synthesized dialogues spanning multiple age groups, Fisher contains real human telephone conversations with substantial background noise, and MultiDialog includes clean, face-to-face dialogue recordings from native speakers.
>
> Given that our model performs both binary classification (human vs. machine) and five-degree rating on multiple human-likeness dimensions, we evaluated the generalization from these two perspectives.
>
> **(i) Human-machine discrimination**
> We calculated the Pearson correlation coefficient between the model’s binary predictions and human annotations, along with the classification accuracy. As shown in the table below, the judge model exhibits strong correlation with human and high accuracy on OOD data, demonstrating strong generalization beyond the training distribution.
>
> | Dataset | Ours (In-Domain) | Out-of-Domain |
> | --- | --- | --- |
> | Pearson r | 0.9012 | 0.9658 |
> | Accuracy | 0.9605 | 0.9740 |
>
> **(ii) Per-dimension rating using five degrees**
>
> To evaluate generalization for the five-degree rating, we invited human experts to annotate the OOD samples on multiple dimensions. As shown in the table below, we report Pearson correlation for several representative dimensions and all dimensions. The results confirm that our model also generalizes well at the fine-grained level.
>
> | Dataset | Ours (In-Domain) | Out-of-Domain |
> | --- | --- | --- |
> | Auxiliary Vocalizations | 0.6292 | 0.7329 |
> | Micro-physiological Noise | 0.7478 | 0.7198 |
> | Pronunciation Instability | 0.8071 | 0.7548 |
> | all dimensions | 0.6284 | 0.6728 |
>
> [3] Du, Zhihao, et al. Cosyvoice 2: Scalable streaming speech synthesis with large language models. arXiv preprint arXiv:2412.10117 (2024).
>
> [4] Cieri, Christopher, David Miller, and Kevin Walker. The Fisher corpus: A resource for the next generations of speech-to-text. LREC. 2004.
>
> [5] Park, Se, et al. Let’s Go Real Talk: Spoken Dialogue Model for Face-to-Face Conversation. ACL 2024.

---

> ### Author Response · Authors · 2025-11-19
> **Replies to Reviewer 3UQQ: second part**
>
> ####  **Weakness 3: The contradiction  in Q2  between line 278 and line 332**
>
> We apologize for the confusion caused by our wording.
> The two statements are indeed not contradictory. It looks contradictory because the  **context** of the two statements are omitted.
> - Context of Observation 1 (line 278-279)  :  at the $\color{blue} \textrm{vocal}$  level (e.g. audio and speech) with a S2S protocol.
> - Context of Observation 3 (line 332) : at the $\color{red}  \textrm{text}$ level
>
> We rephrased them as below:
>
> For the statement in Observation 1:
> > However, their performance surpasses that of most S2S systems, revealing that today's S2S systems are limited not only by vocal quality, but also by ${\color{blue} \textrm{vocal}}$ interaction capabilities such as  ~~topic~~   ${\color{blue} \textrm{speech}}$ understanding,$ {\color{blue} \textrm{role-based acoustic}}$  adherence, and conversational reasoning.
>
> For the statement in Observation 3:
> > These strengths indicate that S2S systems have largely solved the foundational challenges of $\color{red} \textrm{textual}$ understanding and generating clear and coherent dialogue ~~turns~~  $\color{red} \textrm{scripts}$.
>
>
>
> Therefore, we revise it using blue color, see the updated version.
>
> ####  **Q1: Language proficiency of participants**
>
> Not all participants speak both Chinese and English. In fact, 4 out of 28 participants recorded dialogues in both languages. The remaining participants recorded in only one language: 14 participants recorded only English, and 10 participants recorded only Chinese. We provide the detailed language distribution of all 28 participants in the table below, and we will include this information in the revised version for clarity.
>
>
> | speakerid | Chinese | English | Country / Region        |
> |-----------|---------|---------|--------------------------|
> | speaker01 | ✔       | ✘       | China                    |
> | speaker02 | ✔       | ✔       | China                    |
> | speaker03 | ✔       | ✘       | China                    |
> | speaker04 | ✔       | ✘       | China                    |
> | speaker05 | ✔       | ✔       | China                    |
> | speaker06 | ✔       | ✔       | China                    |
> | speaker07 | ✔       | ✘       | China                    |
> | speaker08 | ✘       | ✔       | China                    |
> | speaker09 | ✔       | ✘       | China                    |
> | speaker10 | ✔       | ✘       | China                    |
> | speaker11 | ✔       | ✘       | China                    |
> | speaker12 | ✔       | ✘       | China                    |
> | speaker13 | ✔       | ✔       | China Hong Kong    |
> | speaker14 | ✘       | ✔       | Pakistan                 |
> | speaker15 | ✘       | ✔       | Tajikistan               |
> | speaker16 | ✘       | ✔       | Malaysia                 |
> | speaker17 | ✘       | ✔       | Indonesia                |
> | speaker18 | ✘       | ✔       | Russia                   |
> | speaker19 | ✘       | ✔       | Indonesia                |
> | speaker20 | ✘       | ✔       | Greece                   |
> | speaker21 | ✘       | ✔       | Indonesia                |
> | speaker22 | ✘       | ✔       | Indonesia                |
> | speaker23 | ✘       | ✔       | UK                       |
> | speaker24 | ✘       | ✔       | US                       |
> | speaker25 | ✘       | ✔       | Indonesia                |
> | speaker26 | ✔       | ✘       | China                    |
> | speaker27 | ✔       | ✘       | China                    |
> | speaker28 | ✘       | ✔       | Indonesia                |
>
>
>
> ####  **Q2**
> Please refer to response to the **Weakness 1**.

---

> ### Comment · Reviewer_3UQQ · 2025-11-24
>
> Thank you for your response. I found the comparisons to VoiceBench and MMAU are very informative, and I think they further highlight the value of the proposed test. Also, it is good to see the evaluation results correlate well with humans. Also, thanks for updating the discussions, I now understand there is no contradictions of the takeaways.
>
> Overall, the review addressed my concerns, and I updated my score (6).

---

> > ### Author Response · Authors · 2025-11-24
> >
> > Thank you for your positive feedback and for increasing the score. We are pleased to have addressed your concerns and are grateful for your valuable input. We will incorporate the points discussed above into the revised version to further enhance the clarity of the paper.

---

### Official Review · Reviewer_tKss · 2025-10-30

**Soundness:** 3
**Presentation:** 4
**Contribution:** 3
**Rating:** 8
**Confidence:** 3

**Summary:**

This paper introduces a Turing test study for speech-to-speech (S2S) dialogue systems, evaluating 9 LLMs against human speakers. Using a gamified online platform, the authors collect 2,968 human judgments across 1,486 dialogues in English and Chinese. None of the tested systems passed the Turing test, revealing a persistent gap in human-likeness. To understand the causes, the paper develops a taxonomy of 18 human-likeness dimensions, spanning semantic, paralinguistic, emotional, and persona-related traits. Crowd annotations show that while S2S systems perform near human levels in semantic coherence, they fall short in prosody, emotional expression, and conversational naturalness. Finally, the paper proposes an interpretable AI judge, which is a finetuned LLM that predicts human-likeness with strong transparency and accuracy, outperforming both humans and baseline AI judges, either prompted or LoRA-finetuned.

**Strengths:**

- This paper presents the first formal Turing test for S2S dialogue systems, extending evaluation beyond text to spoken interaction, which is an impactful direction given recent advances in conversational AI.
- The paper convincingly shows that the bottleneck of S2S dialogue systems is no longer semantic understanding but rather paralinguistic and emotional expressivity, which is an under-explored dimension in S2S research, offering valuable insights for improving S2S design.
- The interpretable AI judge is a standout contribution, which provides a reproducible and scalable framework for automatic S2S evaluation, with decent human-machine discrimination accuracy.

**Weaknesses:**

- The gamified Turing test platform may attract casual participants who do not conduct the human-machine discrimination carefully. This paper does not clearly describe participants’ quality-control mechanisms  such as attention checks, response-time filtering, etc. This could bias the Turing test results.
- It is unclear how many unpassed Turing test cases are because LLMs avoid human disfluency cues (or other fixes of human speech deficiencies). This is an easy-to-detect feature but a minor issue, since superior speech fluency is rather preferred by users in real-time applications. It would be better to track the cause of each Turing test failure and analyze the rate of such minor causes versus more severe causes.

**Questions:**

- Any additional discussions or experimental results to resolve the above weaknesses?
- Would shorter dialogues (e.g., 20-second versus 60-second dialogues) have more chance to pass the Turing test?

---

> ### Author Response · Authors · 2025-11-19
> **Replies to Reviewer tKss**
>
> We deeply thank Reviewer tKss for the positive assessment of our Turing test for S2S dialogue systems, the empirical insights presented in the paper, and the contribution of our interpretable AI judge. We address each comment in detail below.
>
> #### **Weakness 1: Quality Control in the Turing Test**
>
> In our experimental design, we have implemented a quality assurance mechanism using "**trap audios**" to detect and filter out casual or inattentive responses. Specifically, during each round where participants evaluate five audio clips, there is a 30% probability that one of the clips will be a trap audio—a deliberately synthesized robotic voice repeating the phrase "I'm AI." We expect that listeners can easily recognize this clip as machine-generated. If a participant incorrectly identifies the trap audio, the entire round of data is discarded. If correctly identified, responses to trap audios are excluded from the final Turing test results.
>
> We will make this quality-control procedure clearly stated in the revised version. Thank you again for your valuable feedback.
>
> ####  **Weakness 2: Failure-cause analysis**
>
> Thank you for this insightful suggestion. To better understand the causes of Turing-test failures, we conducted a supplementary small-scale survey within the limited time available for rebuttal. We evenly partitioned the original Turing-test dataset into ten subsets of human–human (H-H) dialogues and ten subsets of human–machine (H-M) dialogues, and invited 10 human judges (Master’s/PhD level, with substantial experience using AI systems). Each judge was assigned one H-H subset and one H-M subset, which were merged and randomly shuffled into a single evaluation set. For each dialogue, judges first performed the same human/AI discrimination task as in our main study. If a dialogue was judged as AI, a follow-up question was asked: “Was your decision mainly because the speech was too fluent or lacked human-like disfluency?”
>
> In this survey, the accuracy for H-H dialogues was 71.8% (102/142), and for H-M dialogues 86.1% (124/144). Among the 124 H-M dialogues that failed the Turing test (i.e., correctly identified as machine), only 14.5% (18/124) were attributed primarily to “overly fluent” speech or lack of human disfluency. This suggests that disfluency-related cues represent only a minor share of failure cases, and that most failures are instead driven by more substantial deficits in human-likeness.
>
> In addition, we examined how much our model relies on disfluency-related cues when identifying machine-generated speech. Among the 18 diagnostic dimensions, we selected three that are most conceptually linked to human disfluency—Linguistic Imprecision, Use of Fillers, and Rhythm. For all H-M dialogues that our model classified as machine (i.e., failed the Turing test), we computed each dimension’s contribution, defined as the product of the ODL output logits and the corresponding linear weight in the FL layer. As shown in the table below, all three dimensions exhibit negative contributions, meaning they push the model toward predicting “machine.” However, their combined contribution accounts for only 20.1% of the total negative contribution, suggesting that disfluency plays only a relatively minor role in the model’s machine-identification decisions.
>
> | Dimension | Contribution |
> | --- | --- |
> | Linguistic Imprecision | -0.9456 |
> | Use of Fillers | -1.2126 |
> | Rhythm | -0.8214 |
> | **Overall (sum of all negative contributions)** | **-14.8243** |
>
> Taken together, both results consistently suggest that missing disfluency cues are not a major cause of Turing-test failures—either for human judges or our automated AI judge. We sincerely appreciate the reviewer for raising this important point. Your suggestion has been extremely valuable in strengthening our study. We will incorporate more fine-grained cause options in future Turing-test designs to better characterize the sources of failure.

---

> ### Author Response · Authors · 2025-11-19
> **Replies to Reviewer tKss: second part**
>
> #### **Q1**
> Please refer to our responses to Weaknesses 1 and 2.
>
> #### **Q2: Influence of Dialogue Length on Turing Test Performance**
> We divided the Turing test results by dialogue length and calculated the classification accuracy for different dialogue types: human-human dialogues (H-H), human-machine dialogues (H-M), and pseudo-human dialogues (PH). The table below summarizes the accuracy results across different duration ranges:
>
> | Duration |  H-H (acc/count) | H-M (acc/count) | PH (acc/count) |
> | --- | --- | --- | --- |
> | [20,25) | 0.4000 / 5 | N/A / 0 | 0.6624 / 157 |
> | [25,30) | 0.7800 / 50 | 0.7742 / 31 | 0.6654 / 257 |
> | [30,35) | 0.6513 / 152 | 0.8333 / 126 | 0.6337 / 243 |
> | [35,40) | 0.7033 / 246 | 0.8642 / 162 | 0.6087 / 92 |
> | [40,45) | 0.6839 / 174 | 0.8498 / 273 | 0.6200 / 50 |
> | [45,50) | 0.7179 / 78 | 0.8564 / 195 | 0.6333 / 60 |
> | [50,55) | 0.8421 / 76 | 0.7737 / 137 | 0.5349 / 43 |
> | [55,60) | 0.7234 / 141 | 0.7907 / 43 | N/A / 0 |
>
> We performed Cochran–Armitage Trend Tests to examine the potential linear relationship between dialogue length and accuracy and found no significant trend for any individual dialogue type. This suggests that dialogue length alone does not significantly influence the likelihood of passing the Turing test.
>
> | Dialogue Type | Z Statistic | p-value | Significant Trend? |
> | --- | --- | --- | --- |
> |  H-H  | 1.6604 | 0.09683 | ✘ |
> | H-M | -1.0106 | 0.31220 | ✘ |
> | PH | -1.6018 | 0.10919 | ✘ |

---

> ### Comment · Reviewer_tKss · 2025-11-24
>
> Thank you for your response. I maintain my positive rating.

---

> > ### Author Response · Authors · 2025-11-24
> >
> > Thank you for your positive rating and for the time you devoted to reviewing our manuscript. Your constructive feedback has been invaluable in helping us further strengthen the work. We will incorporate all of your suggestions into the revised version.

---

### Official Review · Reviewer_zvPu · 2025-11-01

**Soundness:** 3
**Presentation:** 3
**Contribution:** 2
**Rating:** 4
**Confidence:** 4

**Summary:**

The paper targets speech‑to‑speech (S2S) dialogue and proposes a Turing‑style evaluation. It builds a dataset that includes human–human, human–machine, and pseudo‑human (TTS‑synthesized) conversations, and uses a game‑based human study to judge whether current systems “sound human.” The headline finding is that none of the evaluated systems pass. To explain why, the authors introduce a fine‑grained human‑likeness taxonomy and crowd annotations, showing that shortcomings lie in paralinguistics (rhythm, intonation, stress, fillers, breath), emotional expressivity, and a “mechanical persona,” rather than semantics. Off‑the‑shelf AI judges are unreliable; authors therefore propose an interpretable evaluator that first produces ordinal scores on the taxonomy dimensions and then makes a transparent linear human‑vs‑machine decision.

**Strengths:**

1. Focused problem and clear protocol. The work targets the central question for speech‑to‑speech (S2S): Do these systems actually sound human in multi‑turn dialogue? Instead of testing isolated sub‑skills, the study frames evaluation as a Turing‑style decision under realistic interaction. The tri‑part setup, human–human, human–machine, and a TTS‑based pseudo‑human control, gives a clean yardstick for what “human‑like” means. Bilingual coverage and multiple everyday topics reduce overfitting to any single style and make results more comparable across systems. The recording and interaction procedures are standardized, improving internal validity and making cross‑system contrasts meaningful.
2. Interpretable automatic judge. The proposed evaluator is intentionally two‑stage: first map dialogs to ordinal scores on the human‑likeness dimensions, then apply a linear decision with symmetry regularization. This keeps the prediction space aligned with how humans actually rate speech (ordered categories), while the linear head provides transparent attribution: which dimensions pushed a sample toward “human” vs. “machine.” The design is modular, portable across collections, and produces diagnostics that engineers can act on (e.g., prosody shaping, disfluency modeling, persona calibration).
3.Memorable headline result. The paper lands a crisp, communicable takeaway: contemporary S2S systems still fail a Turing‑style test. That single sentence is easy for the community to remember and cite, and it reframes progress: sounding human is not simply a by‑product of better recognition or text generation. Because the result was obtained under a matched protocol with both human–human and synthesized controls, it carries weight beyond a one‑off demo and can serve as a reference point for future work.

**Weaknesses:**

1. Application‑heavy, limited theoretical novelty. The main novelty lies in system integration rather than theory. The core claim, that semantics alone cannot sustain effective speech interaction, is treated as an empirical observation, not a theoretical insight. Adding paralinguistic cues (prosody, affect, persona) targets known gaps, long discussed in TTS and affective computing. The work validates their importance but does not explain underlying mechanisms or interactions, nor does it offer a general theoretical framework.
2. HCI‑leaning narrative.The main text emphasizes human‑study design, demographics, and logistics more than theory, ablations, and generalization, which may misalign with ICLR.
The manuscript emphasizes HCI logistics (recruitment, demographics, task design) while skimming key ML details (architecture ablations, training dynamics, hyperparameter sensitivity). This balance misaligns with ICLR.
3. Limited external generalization and statistics. Training and evaluation are tightly coupled to the same data collection protocol, raising distribution-shift concerns. Evaluation relies on a single-threshold binary decision without calibration, uncertainty quantification, or decision-boundary analysis. Missing ablations across acoustic conditions, speaker populations, and interaction settings leave generalization in doubt. Statistical reporting should add uncertainty intervals, significance tests, calibration curves, and failure-mode taxonomies to substantiate reliability and scope. Also need more systematic ablations to clarify the theoretical footing of ODL, justify the 18‑dimension design and strengthen label reliability.

**Questions:**

Q1. Theoretical Positioning of ODL

What is the precise formal correspondence between ODL and classical ordinal regression models (cumulative-link, threshold models)? What does ODL add beyond these baselines in parameterization or inductive bias? Which properties are guaranteed by construction versus empirically observed?

Q2. Necessity of 18 Dimensions

Why exactly 18 dimensions rather than a single score or reduced factorized set? Provide dimension-wise correlation analysis, clustering structure, and direct comparisons with (i) single-score model and (ii) low-factor model to demonstrate irreducibility.

Q3. Annotation Reliability and Expert Impact

Need to report inter-rater reliability for dimensions. Quantify how expert edits change label distributions and downstream judge performance. Test measurement invariance across languages and subgroups.

---

> ### Author Response · Authors · 2025-11-19
> **Replies to Reviewer zvPu**
>
> #### **Weakness 1： Application‑heavy, limited theoretical novelty. HCI‑leaning narrative.**
>
> We acknowledge that our paper is application-heavy; however, the ICLR community does not exclude application-oriented work. In fact, the proportion of such papers has been steadily increasing.
>
> Although this is an empirical paper, we do have *theoretical grounding*. Specifically, we introduce 18 human-likeness dimensions that form a fine-grained evaluation taxonomy, inspired by *theories* grounded in established cognitive and social science literature (Table 6). To the best of our knowledge, this is the first human-likeness taxonomy for speech interaction, providing a principled foundation for future research on human-like S2S behavior.
>
>
> ##### **Weakness 2： HCI‑leaning narrative**
>
> This paper is an empirical study, and providing clear descriptions of our recruitment procedure, participant demographics, and task design helps ensure proper understanding and reproducibility of our quantitative results. Based on your suggestions, we have moved some of these details to the appendix, which allowed us to free up space in the main paper to include additional experimental results (e.g., relations between dimensions, correlation analyses, and OOD generalization tests).
>
> Presenting such implementation (e.g. recruitment procedure, participant demographics, and task design) is a matter of scientific rigor for empirical work; we believe that it is independent of whether the paper is positioned as an HCI paper, an ICLR theory/application paper. As noted above, we hope that reducing the proportion of these details in the main text can help alleviate your concerns.
>
> #### **Weakness 3：Limited external generalization and statistics.**
>
> Our submission includes an OOD evaluation in Table 3 on the Pseudo-Human test set. We will clarify this more explicitly in the revised manuscript.
>
> We further evaluated our model on three new datasets designed to cover diverse acoustic, demographic, and interaction conditions:
> - CosyVoice2 Synthesis [1] (Pseudo Human): Synthesized dialogues across different age groups (older adults and children).
> - Fisher [2] (Human-Human): Telephone speech with significant background noise.
> - MultiDialog [3] (Human-Human): Clean background native-speaker dialogue recordings.
>
> We sampled 64 dialogues from each dataset for evaluation. In addition to accuracy, we introduced the ROC-AUC score to provide a robust and threshold-independent evaluation of classification performance. The results of human–machine classification are presented in the table below. These results indicate that the model generalizes well and maintains stable performance under distribution shift.
>
> | Data Type | Ours | CosyVoice2 | Fisher | MultiDialog | OOD Combined (CosyVoice2 + Fisher + MultiDialog) |
> | --- | --- | --- | --- | --- | --- |
> | Our Model (Acc) | 0.9605 | 0.9844 | 0.9844 | 0.9531 | 0.9740 |
> | Our Model (ROC AUC) | 0.9791 | — | — | — | 0.9881 |
>
> To evaluate the model’s generalization for the five-degree rating, we invited human experts to annotate the OOD samples on multiple dimensions and report three accuracy metrics, where Exact is the percentage of predictions that exactly match the expert score, Group is the percentage that fall into the same human–machine identity group (1–2 machine-like, 3 unclear, 4–5 human-like), and Nearby is the percentage that differ from the expert score by at most ±1. The results are shown in the table below, indicating that our model maintains strong generalization ability in fine-grained scoring.
>
> | Dataset | Exact | Group | Nearby |
> | --- | --- | --- | --- |
> | Ours | 0.7056 | 0.8408 | 0.8470 |
> | CosyVoice2 | 0.6450 | 0.7569 | 0.8030 |
> | Fisher  | 0.6476 | 0.7396 | 0.7752 |
> | MultiDialog | 0.6562 | 0.7561 | 0.7847 |
>
> We also computed the quadratic weighted Cohen’s Kappa κ between the expert and the model on the OOD dataset to assess their consistency. The resulting κ = 0.6645 indicates that the experts’ and the model’s fine-grained scores exhibit a substantial level of agreement on OOD data, which reflects generalization at a fine-grained level.
>
> [1] Du, Zhihao, et al. Cosyvoice 2: Scalable streaming speech synthesis with large language models. arXiv preprint arXiv:2412.10117 (2024).
>
> [2] Cieri, Christopher, David Miller, and Kevin Walker. The Fisher corpus: A resource for the next generations of speech-to-text. LREC. 2004.
>
> [3] Park, Se, et al. Let’s Go Real Talk: Spoken Dialogue Model for Face-to-Face Conversation. ACL 2024.

---

> ### Author Response · Authors · 2025-11-19
> **Replies to Reviewer zvPu: second part**
>
> #### **Q1. Theoretical Positioning of ODL**
>
> Thank you for the question. We would like to clarify that the ODL is not a core contribution of our method, but rather a practical choice for modeling the inherently ordered 1–5 human-likeness scores. The ODL is conceptually related to classical cumulative-link ordinal regression models, in that both use cumulative probabilities to encode ordinal structure. We adopt ODL because it provides a lightweight and differentiable way to enforce ordinal constraints through monotonic cumulative probabilities and learnable cut-points, without requiring a full probabilistic ordinal-regression formulation. This makes it easy to integrate into modern neural architectures while still respecting the ordered nature of the labels.
>
> To validate the effectiveness of ODL, we conducted an ablation where we removed the ODL and replaced it with a standard linear layer and negative log-likelihood loss, treating the human-likeness scores as independent categories. This baseline corresponds to a non-ordinal but still interpretable classifier.
>
> | Datatype | Human-Human | Human-Machine | Pseudo Human | Overall Acc |
> | --- | --- | --- | --- | --- |
> | Our Model (Ordinal) | 0.9507 | 0.9722 | 0.9306  | 0.9605 |
> | Linear Loss (Non-Ordinal) | 0.8718 | 0.9875 | 0.9097 | 0.9233 |
>
> We hope this clarifies that ODL is used as an appropriate modeling choice for ordinal labels, and that our ablation demonstrates its empirical value.
>
> #### **Q2. Necessity of 18 Dimensions**
>
> **1）Dimension-wise Correlation Analysis**
>
> Thank you for your suggestion. The correlations between dimensions and the question of redundancy are important considerations in constructing evaluation factors. While using basic dimensions such as pitch and rhythm can ensure orthogonality, it may overlook critical human-like features, such as written-style expression, which are harder to capture. Focusing solely on highly critical dimensions risks creating composite factors that may lead to overlap and redundancy. In our approach, we aim to strike a balance by incorporating both foundational aspects, like rhythm, and distinctive features, such as written-style expression.
>
> Based on our annotated data, we computed the correlation matrix across all 18 dimensions, shown below. Due to space constraints, we present only a subset of the results here; the full results will be included in the updated PDF. It’s important to note that these correlation values are indicative rather than conclusive due to the diverse nature of the audio samples (different models, human speakers, and varied dialogue content). The correspondence between the abbreviated dimension names and their full forms can be found in Table 11 of the paper.  It can be observed that most of the correlation values are relatively low, which demonstrates that the dimensions we selected are distinct and capture different aspects of human-likeness.
>
> | Dimension | MC | LC | PA | CS | LI | UF | MM | RT | IT | ST | AV | MN | PI | AC | SB | WE | TS | AE |
> |-----------|----|----|----|----|----|----|----|----|----|----|----|----|----|----|----|----|----|----|
> | Memory Consistency | 1.00 | 0.62 | 0.31 | 0.08 | 0.28 | 0.30 | 0.11 | 0.28 | 0.29 | 0.25 | 0.23 | 0.25 | 0.26 | 0.08 | 0.29 | 0.28 | 0.27 | 0.26 |
> | Logical Coherence  | 0.62 | 1.00 | 0.40 | 0.09 | 0.30 | 0.33 | 0.16 | 0.30 | 0.33 | 0.25 | 0.23 | 0.24 | 0.27 | 0.12 | 0.29 | 0.32 | 0.34 | 0.29 |
>
>
> **2）Dimension Ablation**
>
> We further conducted an ablation study by removing the “Pronunciation Instability” and “Written-style Expression” dimensions separately and retraining the model. The accuracy results are summarized in the table below. The consistent performance degradation indicates that both dimensions contribute essential to modeling human-likeness, supporting the necessity of retaining them in the full 18-dimension framework.
>
> | Dataset | Our model (with full dimensions) | w/o Pronunciation Instability | w/o Written-style Expression |
> | --- | --- | --- | --- |
> | Human-Human | 0.9507 | 0.8803 | 0.8944 |
> | Human-Machine | 0.9722 | 0.9861 | 0.9861 |
> | Pseudo Human | 0.9306 | 0.9167 | 0.9098 |
> | Avg | 0.9605 | 0.9279 | 0.932 |
> | ROC AUC | 0.9791 | 0.9762 | 0.9700 |

---

> ### Author Response · Authors · 2025-11-19
> **Replies to Reviewer zvPu: third part**
>
> #### **Q3. Annotation Reliability and Expert Impact**
>
> Thank you for raising concerns about the reliability of annotations. While the initial labels were obtained through crowdsourcing, their final quality is ensured through expert verification and correction. As presented in the table below, experts revised 29.44% of the labels, with an average adjustment of 1.99 points (39.81% of the score range), demonstrating that expert review effectively removes substantial noise in the raw annotations. Note that Change_Ratio refers to the proportion of adjusted annotations. RMSE and RMSE_Ratio are calculated only on the annotations that were changed. Due to space limitations, we present only the three dimensions with the highest Change_Ratio, along with the overall results across all 18 dimensions.
>
> | Dimension | Change_Ratio | RMSE | RMSE_Ratio |
> | --- | --- | --- | --- |
> | Pronunciation Accuracy | 0.3596 | 2.1085 | 0.5271 |
> | Textual Sentiment | 0.3472 | 1.9579 | 0.4895 |
> | Linguistic Imprecision | 0.3273 | 2.1230 | 0.5308 |
> | Overall | 0.2944 | 1.9903 | 0.4976 |
>
> To further validate annotation reliability, we trained models on data before and after expert correction. Expert-refined labels lead to substantial improvements in both in-distribution and OOD generalization, confirming the quality of our final annotation set.
>
> |                      | Ours (In-Domain) | CosyVoice2| Fisher | MultiDialog | Avg (OOD) |
> |----------------------|----------------------|-----------------|------------|------------------|----------------|
> | Before           | 0.8791               | 0.9375          | 0.6250     | 0.9062           | 0.8229         |
> | After           | 0.9605               | 0.9844          | 0.9844     | 0.9531           | 0.9740         |
>
>
>
> We conducted a cross-lingual consistency analysis comparing the English and Chinese annotations across all 18 fine-grained dimensions. As shown in the below table, the mean and variance of each dimension across the two languages closely align, and the overall RMSE between English and Chinese scores is 0.24, which accounts for only 6% of the full scoring range. This indicates that the fine-grained scoring scheme is interpreted consistently across languages.
>
> | Category | Mean (EN) | Variance (EN) | Mean (CN) | Variance (CN) | CN−EN (Mean Diff) |
> | --- | --- | --- | --- | --- | --- |
> | Pronunciation Accuracy | 4.3327 | 1.2718 | 4.4157 | 1.1273 | 0.0830 |
> | Textual Sentiment | 3.3051 | 2.5359 | 3.5997 | 2.1244 | 0.2946 |
> | Linguistic Imprecision | 3.0413  | 2.8602 | 3.5417 | 2.2862 | 0.5004 |
> | Overall | 3.3891 | 2.3110 | 3.5577 | 2.2592 | 0.1686 |
>
> #### **Hyperparameter sensitivity analysis**
>
> As a complementary experiment to our main hyperparameter tuning, we performed a 1000-run randomized hyperparameter search, sampling key training parameters for ODL (learning rate, batch size, scale, dropout) and FL (learning rate, batch size). Each configuration was trained end-to-end using the same evaluation protocol, ensuring reliability through full parallelization. The results for the hyperparameter sensitivity analysis (accuracy) are presented in the tables below.
>
> | Hyperparameter | Values | ODL Val Acc Mean ± Std | FL Test Acc Mean ± Std | Range (ODL) | Range (FL) | Rel-Var% (ODL) | Rel-Var% (FL) | MSE (ODL) | MSE (FL) |
> | --- | --- | --- | --- | --- | --- | --- | --- | --- | --- |
> | odl_lr | {1e-05, 1e-04, 1e-03, 1e-02} | 0.6020 ± 0.0435 | 0.8642 ± 0.0071 | 0.0930 | 0.0166 | 7.2259 | 0.8216 | 0.002174 | 0.000051 |
> | odl_batch_size | {32, 64, 128, 256} | 0.6105 ± 0.0065 | 0.8601 ± 0.0149 | 0.0154 | 0.0360 | 1.0647 | 1.7324 | 0.000048 | 0.000222 |
> | scale | {1, 1.05, …, 5} | 0.6293 ± 0.0090 | 0.9254 ± 0.0283 | 0.0533 | 0.1724 | 1.4078 | 3.0581 | 0.000091 | 0.000802 |
> | dropout | {0.1, 0.2, 0.3, 0.4, 0.5} | 0.6103 ± 0.0050 | 0.8617 ± 0.0103 | 0.0134 | 0.0268 | 0.8193 | 1.1953 | 0.000029 | 0.000105 |
> | fl_lr | {1e-05, 1e-04, 1e-03, 1e-02} | 0.6109 ± 0.0031 | 0.8584 ± 0.0696 | 0.0072 | 0.1527 | 0.5074 | 8.1081 | 0.000011 | 0.004838 |
> | fl_batch_size | {16, 32, 64, 128} | 0.6107 ± 0.0034 | 0.8650 ± 0.0193 | 0.0080 | 0.0435 | 0.5567 | 2.2312 | 0.000013 | 0.000371 |
>
> Analyzing the results, we identify several key findings:
>
> - Learning rate proved to be a critical factor for both ODL and FL, consistent with findings from other work. Extremes caused underfitting or instability, emphasizing the need for precise tuning.
> - Scale had minimal impact on ODL accuracy, suggesting ODL’s adaptability, but slightly affected FL due to scale-induced changes in logits cut-points.
> - Batch size influenced FL performance, with larger batches stabilizing training but potentially slowing convergence or causing overfitting.
> - Dropout and ODL batch size showed minimal effects, indicating that ODL is robust to these parameters.
>
> Overall, the 1000-run analysis shows that our method is generally robust, with learning rate being the most sensitive parameter, while other hyperparameters produce only modest effects.

---

> ### Comment · Reviewer_zvPu · 2025-11-20
>
> Thank you to the authors for the detailed response and the additional experiments, which have addressed many of my concerns. I hope you can incorporate these experiments and the necessary discussions into the revised version of the paper.
> I will increase my score by 2 points. I look forward to seeing this paper published soon.

---

> ### Author Response · Authors · 2025-11-20
>
> Thank you very much for your decision to increase the score. We are glad that our response addressed your concerns. We truly appreciate the time and effort you dedicated to reviewing our work, and your comments have been very helpful in improving the paper. We will incorporate these experiments and discussions into the revised version.

---

### Official Review · Reviewer_pdbT · 2025-11-02

**Soundness:** 4
**Presentation:** 4
**Contribution:** 4
**Rating:** 8
**Confidence:** 5

**Summary:**

This paper performs a Turing test experiment with multiple speech to speech models. Multiple versions of the experiment are tested using controlled conditions that attempt to minimize potential confounds. To collect more data, a mobile app is released as well. Conversations were also annotated with 18 dimensions. The results show key gaps in all S2S models. Using these insights, a smaller speech model is fine-tuned to predict these ratings and then classify whether a given dialog is human-to-human or human-to-s2s, attaining very high accuracy while still being interpretable.

**Strengths:**

- Clean experimental setup that controls for multiple confounds and tests a diverse suit of S2S models, including TTS models using LLM-generated text. Experiments demonstrate key gaps.

- Rich analysis of why models fail the test through a multifaceted analysis of the conversations qualities. This analysis involved collecting human perceptions using crowdsourcing

- Experiments to see if other audio models could pass the test, with key gaps in existing models. Proposes new model and design to get an interpretable explanation, showing that these features (when correctly identified) are reliable indicators of gaps in current system

- Released platform and game to collect new annotations, helping grow data and potentially incorporate more models in the future

- Very well written paper with clear motivation, analysis, and visuals.

**Weaknesses:**

- The biggest gap to me was in the lack of details around the annotation for conversation qualities. These are barely mentioned in text, so I was expecting to see a much more detailed report in B.5. However, important questions are hard to answer, such as who annotated (which platform?), how many annotators were there, did annotators agree on these qualities, how much were annotators paid, or what quality controls were present, if any. Given the importance of this data for your results and later test-taking model, more details are needed to assess the quality of the data and for future replicability.

- The paper itself is very dense (though well written). However, the space constraint has pushed many details to the appendix which hinders readability at times.

**Questions:**

- How was the annotation performed (see questions above)

---

> ### Author Response · Authors · 2025-11-19
> **Replies to Reviewer pdbT**
>
> We sincerely appreciate your recognition of our work, including the experiments, analyses, the game platform, and the writing. We address your concerns and questions below, and the corresponding details have been added to Section 5 and Appendix B.5 of the rebuttal version.
>
> #### **Weakness 1： lack of details around the annotation for conversation qualities**
> - **Annotation Platform**
>
> As shown in Figure 13 of our paper, we deployed an annotation platform on Hugging Face Space to collect, manage, and analyze annotators’ submissions. For each item, annotators were presented with an audio clip of a dialogue and were asked to listen to it carefully and rate 18 dimensions on a 1–5 scale (with higher scores indicating stronger human-likeness on that dimension). Due to anonymization requirements during the review process, we are unable to publicly release the platform at this stage. We will include the platform link in the final version once the paper is accepted.
>
> - **Annotator Details**
>
> A total of 36 annotators participated in our study. They are master’s and Ph.D. students with backgrounds in AI, and have strong proficiency in both English and Chinese. Before beginning the scoring task, each annotator was required to read the detailed annotation guidelines (Figure 12). They subsequently completed several trial batches, each containing 5 items and requiring approximately 20–30 minutes to finish. Annotators were compensated at a rate of 30 units/hour (local currency), with a total cost equivalent to approximately 5,250 units.
>
> - **Annotation Quality Control**
>
> For quality control, we invited three experts specializing in human-computer interaction research to conduct cross-validation on all submitted content. The experts were given access to the true labels indicating whether each dialogue was responsed by a human or a machine. All annotations were cross-checked by the three experts. Only annotations that were unanimously approved by all three experts were directly included in the dataset. Any disapproval from a single expert led to the data being revised through expert discussion. As presented in the table below, experts revised 29.44% of the labels, with an average adjustment of 1.99 points (49.76% of the score range), demonstrating that expert review effectively removes substantial noise in the raw annotations. Note that Change_Ratio refers to the proportion of adjusted annotations. RMSE and RMSE_Ratio are calculated only on the annotations that were changed. Due to space limitations, we present only the three dimensions with the highest Change_Ratio, along with the overall results across all 18 dimensions.
>
> | Dimension | Change_Ratio | RMSE | RMSE_Ratio |
> | --- | --- | --- | --- |
> | Pronunciation Accuracy | 0.3596 | 2.1085 | 0.5271 |
> | Textual Sentiment | 0.3472 | 1.9579 | 0.4895 |
> | Linguistic Imprecision | 0.3273 | 2.1230 | 0.5308 |
> | Overall | 0.2944 | 1.9903 | 0.4976 |
>
> To further validate annotation reliability, we trained models on data before and after expert correction. Expert-refined labels lead to substantial improvements in both in-distribution and OOD generalization, confirming the quality of our final annotation set.
>
> |                      | Ours (In-Domain) | CosyVoice2 [1]| Fisher [2] | MultiDialog [3] | Avg (OOD) |
> |----------------------|----------------------|-----------------|------------|------------------|----------------|
> | Before           | 0.8791               | 0.9375          | 0.6250     | 0.9062           | 0.8229         |
> | After           | 0.9605               | 0.9844          | 0.9844     | 0.9531           | 0.9740         |
>
> [1] Du, Zhihao, et al. Cosyvoice 2: Scalable streaming speech synthesis with large language models. arXiv preprint arXiv:2412.10117 (2024).
>
> [2] Cieri, Christopher, David Miller, and Kevin Walker. The Fisher corpus: A resource for the next generations of speech-to-text. LREC. 2004.
>
> [3] Park, Se, et al. Let’s Go Real Talk: Spoken Dialogue Model for Face-to-Face Conversation. ACL 2024.
>
> #### **Weakness 2： Readability impacted by details in the appendix.**
>
> Thank you for the comment. The 9-page limit of the initial submission required us to move several explanations to the appendix. For the rebuttal revision, the page limit increases to 10 pages, and we will use this additional space to move key details back into the main text to improve readability. We will upload the updated revision as soon as possible.
>
> #### **Q1**
> Please refer to response to the Weakness 1.

---

### Author Response · Authors · 2025-11-29
**General Response to the Area Chair and Reviewers**

Dear Area Chair and Reviewers,

We sincerely thank all reviewers for their thoughtful and constructive feedback. Below, we summarize the major revisions made in response.

- **Positioning and Contribution**. We clarified how our Turing-test framework offers insights that go beyond existing speech benchmarks such as VoiceBench and MMAU-Pro. Whereas prior benchmarks focus on intelligence-oriented abilities (e.g., understanding, reasoning) and evaluate models through text-based outputs, our framework directly measures **human-likeness in speech-to-speech interaction**, assessing whether a system sounds and behaves like a human in multi-turn spoken dialogue.

- **Generalization of the Judge Model**. We added extensive OOD evaluations using CosyVoice2, Fisher, and MultiDialog datasets, which are entirely unseen during training. The judge model maintains high correlation with human annotations and strong classification accuracy, confirming its robust generalization.


- **Dimension Design and Reliability**. We added dimension-wise correlation analyses and ablation studies, showing that the 18 human-likeness dimensions are both non-redundant and necessary.

- **Annotation Reliability and Expert Impact**. Expert verification substantially improved label quality: experts corrected ~29% of annotations, judge models trained on expert-refined labels showed large gains in both in-distribution and OOD performance.

- **More analysis and ablation studies**. We added (i) an ablation study removing ODL from our judge model and (ii) a hyperparameter sensitivity analysis, demonstrating the effectiveness and robustness of our judge model. We further conducted (iii) a failure-cause analysis of Turing-test and investigated (iv) the influence of dialogue length, finding that disfluency cues explain only a small portion of Turing-test failures and that dialogue length has no significant effect on performance.

- **Clarity and Transparency Improvements**. We (i) clarified previously ambiguous statements, (ii) elaborated on annotation details for conversation qualities, (iii) added a complete speaker–language breakdown for all 28 recording participants, and (iv) further explained our quality-control procedures in the Turing Test. These revisions improve the clarity, transparency, and completeness of our presentation.

We would like to highlight a positive development during the **normal discussion period**: our responses led two reviewers to raise their scores, resulting in a final set of **8/6/8/6** (from 8/4/8/2).
| Reviewer       | Initial Rating | After Discussion | Update Date | Official Comment by Reviewer |
|----------------|---------------|------------------|----------------------------------|------|
| Reviewer pdbT     | 8             | 8                |   —             | —    |
| Reviewer zvPu     | 4             | 6               | 20 Nov 2025                           | Thank you to the authors for the detailed response and the additional experiments, which have addressed many of my concerns. I hope you can incorporate these experiments and the necessary discussions into the revised version of the paper. I will increase my score by 2 points. I look forward to seeing this paper published soon. |
| Reviewer tKss  |  8          |  8              | 24 Nov 2025                 | Thank you for your response. I maintain my positive rating. |
| Reviewer 3UQQ  | 2             | 6                | 24 Nov 2025                      | Thank you for your response. I found the comparisons to VoiceBench and MMAU are very informative, and I think they further highlight the value of the proposed test. Also, it is good to see the evaluation results correlate well with humans. Also, thanks for updating the discussions, I now understand there is no contradictions of the takeaways. Overall, the review addressed my concerns, and I updated my score (6).|


We thank all reviewers again for their valuable comments. We hope that this general summary is helpful for the Area Chair’s assessment of our submission, and we sincerely appreciate your time and consideration.

Sincerely,
The Authors

---

### Meta-Review · Area_Chair_DP9N · 2026-01-07

**Summary:**

Interesting contribution that presents a spoken version of Turing test for speech-to-speech systems. The work collects human judgments on about 3K dialogues, between 9 state-of-the-art S2S systems and 28 human participants and shows the SOTA models fail the test in terms of human-likeness. Their analysis reveals that the failures are mainly due to paralinguistic cues, emotional expressivity, and persona rather than semantics. The work introduces an interpretable model for fine-grained evaluation of human-likeness.

**Reviewer Concerns:**

Reviewers requested extension of analysis and ablation studies, which were included during the rebuttals. Furthermore, there were questions regarding the setup, which were also answered in the rebuttals. One concern that still remains is the limited theoretical novelty.

**Reviewer Scores:**

The reviewer with the rating of 2 would increase their rating (they actually also have a note for that purpose). The reviewer who gave a 4 may stick to their rating.

---

### Decision · Program_Chairs · 2026-01-26

Accept (Poster)